# Recent Progress in Nickel and Silica Containing Catalysts for CO₂ Hydrogenation to CH₄

**Nadiyah Albeladi, Qana A. Alsulami** [ID] **and Katabathini Narasimharao** *[ID]

Chemistry Department, Faculty of Science, King Abdulaziz University, P.O. Box 80203, Jeddah 21589, Saudi Arabia; qalselami@kau.edu.sa (Q.A.A.)
* Correspondence: nkatabathini@kau.edu.sa

**Abstract:** The recent unusual weather changes occurring in different parts of the world are caused by global warming, a consequence of the release of extreme amounts of greenhouse gases into the atmosphere. Carbon dioxide (CO₂) is one of these greenhouse gasses, which can be captured and reused to generate fuel through the methanation process. Nickel- and silica-based catalysts have been recognized as promising catalysts due to their efficiency, availability, and low prices. However, these catalysts suffer from metal sintering at high temperatures. Researchers have achieved remarkable improvements through altering conventional synthesis methods, supports, metal loading amounts, and promoters. The modified routes have enhanced stability and activity while the supports offer large surface areas, dispersion, and strong metal–support interactions. Nickel loading affects the formed structure and catalytic activity, whereas doping causes CO₂ conversion at low temperatures and forms basic sites. This review aims to discuss the CO₂ methanation process over Ni- and SiO₂-based catalysts, in particular the silica-supported Ni metal in previously reported research works and point out directions for potential future work.

**Keywords:** carbon dioxide; methanation; methane production; Ni-based catalysts; silica support

## 1. Introduction

Extended urban areas and industrial activities have led to a dramatic increase in greenhouse gases, such as CO₂, into the atmosphere and caused concerning climate changes. This requires an urgent solution to reduce CO₂ emission and create a clean environment. To achieve this goal, most of the existing research has focused on two methods, improving the validity of using renewable energy sources and implementing carbon capture utilization and storage (CCUS) techniques to preserve the planet [1]. CO₂ is used as a raw material for oxazolidinone formation. This compound is involved in the medicine and drug industry and is synthesized using a simple technique at room temperature [2]. Despite the thermal limitation of the CO₂ methanation reaction, significant improvements have been achieved in transforming CO₂ into valuable single carbon materials, such as carbon monoxide, methane, methanol, and formic acid. The reaction of the reverse water gas shift (RWGS) produces carbon monoxide (CO), while heterogeneous catalysis has been utilized to obtain methanol (CH₃OH) [3]. Thermodynamically, the methanation of CO₂ and CO is quicker than that of other reactions for hydrocarbon production. CO₂ gas is highly stable, and molecule separation is costly [4]. This stability has led to low production due to the low adsorption rate of CO₂ in the catalysts. Full understanding of the complexity of the mechanism of CO₂ conversion into hydrogen fuel requires gaining more information, such as from studying the micro mechanisms of the Sabatier reaction [5]. Mucsi et al. determined a method to quantify carbonyl conjugation by studying the calculated enthalpy of the hydrogenation (ΔH_H₂) of different compounds. The carbonyl attached to conjugative species shows a larger percentage of carbonylicity compared to those of the low- or non-conjugative groups. Compounds

with small carbonylicity are more reactive during the addition reaction in comparison with the high carbonylicity compounds [6]. $CO_2$ conversion to methane requires efficient catalysts to be successful. Recently, this has been the aim of much research that has attempted using different catalyst designs [7]. Converting $CO_2$ into methane fuel has been explored via different reactor systems, such as the thermo-catalytic, thermo-catalytic membrane, plasma catalytic, and photocatalytic systems [4]. Metal–support catalysts are extensively used for $CO_2$ fixation reactions with a variety of metals and catalyst supports. Ni is the ideal metal to use, as it offers reasonably high hydrogenation activity and selectivity [8].

Although different transition metals are suitable for the methanation reaction, supported Ni catalysts are extensively used for $CO_2$ methanation due to their high catalytic activity and cost-effective nature [9], but Ni-based catalysts lose activity quickly during the methanation reaction due to their carbon deposition [7]. However, Ni-based catalysts are the most frequently used catalysts for $CO_2$ hydrogenation despite their lack of activity and stability [10]. The roles of the support, Ni loading, the preparation method, and additives have been investigated to enhance the performance of catalysts. According to the literature, the support might play a role in enhancing metal dispersion and tune the structure of the catalyst surface, which could improve $CO_2$ adsorption and influence the reaction mechanism [9]. Silica ($SiO_2$) is a well-studied support material that is thermally stable and provides a significantly high surface area [11]. It increases the stability of Ni-based catalysts by improving their interaction with the active metal [12]. The acidity of the silica support weakens the interaction with $CO_2$. To improve the interaction, the basicity of the Ni-silica catalyst is enhanced with promoters [8].

To the best of our knowledge, there is no review published in the literature that focuses specifically on Ni-$SiO_2$-based catalysts for the $CO_2$ methanation process. The published reviews discussing Ni-based catalysts in general have focused on Ni-supported catalysts and bimetallic Ni-based catalysts with different supports. In this review, our focus is on the efficacy of different Ni-silica catalysts for $CO_2$ methanation, and a detailed discussion is incorporated related to the influence of the synthesis method, modified supports, Ni-loading, and the dopant on the activity and stability of different Ni-based $SiO_2$ catalysts for $CO_2$ hydrogenation. We summarize the existing literature results by discussing and evaluating the current knowledge in the field of utilizing Ni-based $SiO_2$ catalysts for $CO_2$ methanation. From this, the critical gaps are identified for future work and included in the review.

## 2. $CO_2$ Capture Utilization Storage (CCUS)

The expanding of industrialization worldwide is accompanied by higher levels of greenhouse gases in the atmosphere. Concentrated $CO_2$ emissions in the atmosphere have resulted in lowering the ozone layer, which has caused global warming and climate change. This has affected global agriculture and led to water degradation, health, and economic issues [1]. To solve these problems, scientists have devoted their efforts towards developing existing and new renewable energy systems for carbon capture, storage, and utilization technologies. This is an effective solution for not only inhibiting $CO_2$ from being released into the atmosphere but also producing fuel and valuable chemicals [13].

Figure 1 illustrates the possible ways to limit the release of greenhouse gases into the environment. The first step is to limit the use of fossil fuels by adapting the practical use of green energy sources. Simultaneously, technologies for carbon capture and storage (CCS) and carbon capture and utilization (CCU) should be activated. While CCS processes end with the storage of $CO_2$, which does not solve the main issue, CCU processes involve capturing $CO_2$ and converting it into valuable products [14]. This storage and conversion can be performed using a variety of methods, as shown in Figure 1. Absorption into liquid is a suitable storage strategy for $CO_2$, but it is limited by the high energy demands of regenerating of the solvent. The membrane technique has

a variety of advantages over other CCS technologies, such as applicability in isolated areas, simplicity of maintenance, a low-cost installation, and less chemical and energy requirements [15,16]. Photo-catalytic reduction is initiated through direct sunlight, which is considered a renewable energy source, while electrochemical and plasma technology relies on electricity. The utilization of solar energy is widely acceptable in large-scale applications for generating electricity. The produced energy is effectively stored as fuel rather than battery power. Additionally, chemical storage can be implemented on the current line of infrastructure with a high capability [15]. The advanced technology of water electrolysis (power to gas, P2G) has contributed convenient enhancements in carbon dioxide hydrogenation as a source for green fuel [17]. $CO_2$ reduction, or what is named the Sabatier reaction, is considered an exothermic reaction, as shown in following the equation [8]:

$$CO_2 + 4H_2 \rightarrow CH_4 + 2H_2O \tag{1}$$

$$\Delta H \text{ at } 298 \text{ K} = -165 \text{ kJ/mol}$$

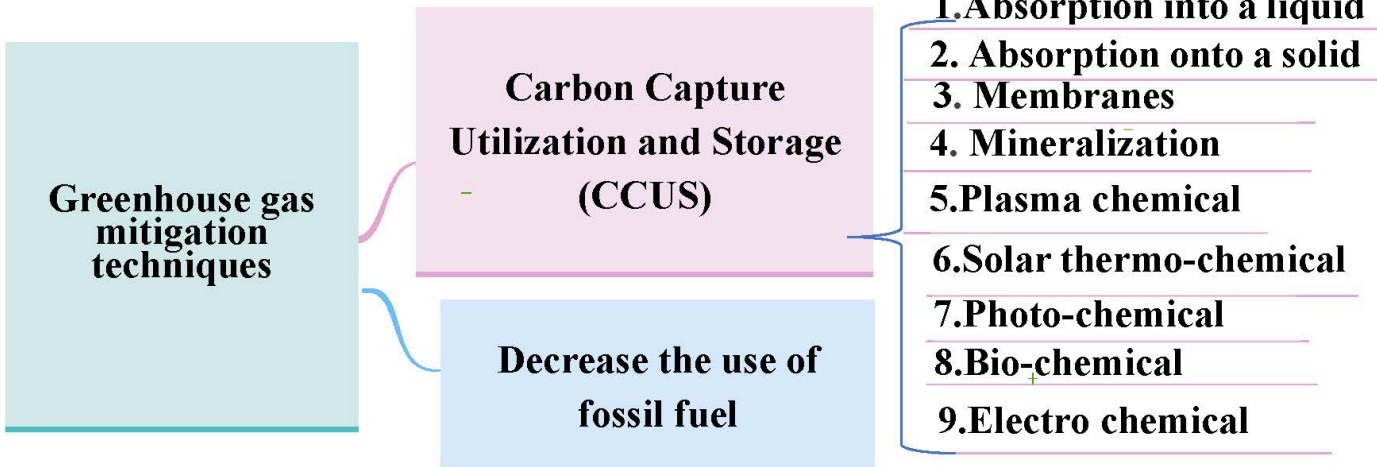

**Figure 1.** Different approaches for $CO_2$ utilization [14,15].

### 3. CO₂ to Valuable Products

Modern heterogeneous catalysts have been developed to convert $CO_2$ into a variety of products with two or more carbon atoms, such as dimethyl ether (DME), olefins, liquid fuels, and higher alcohols. The difficulties of preparing products with multi-carbon atoms, in comparison to single-carbon products, are concentrated in a lack of high $CO_2$ activity and an obstacle in connecting the carbon–carbon bond (C-C) [3].

The introduction of cost-effective approaches has enabled the mitigation of concentrated $CO_2$ by transforming $H_2$ and $CO_2$ into industrially preferable raw materials for producing fuel, in which they are considered a sustainable energy source. Figure 2 presents the different chemicals that can be produced from the carbon dioxide hydrogenation process [15].

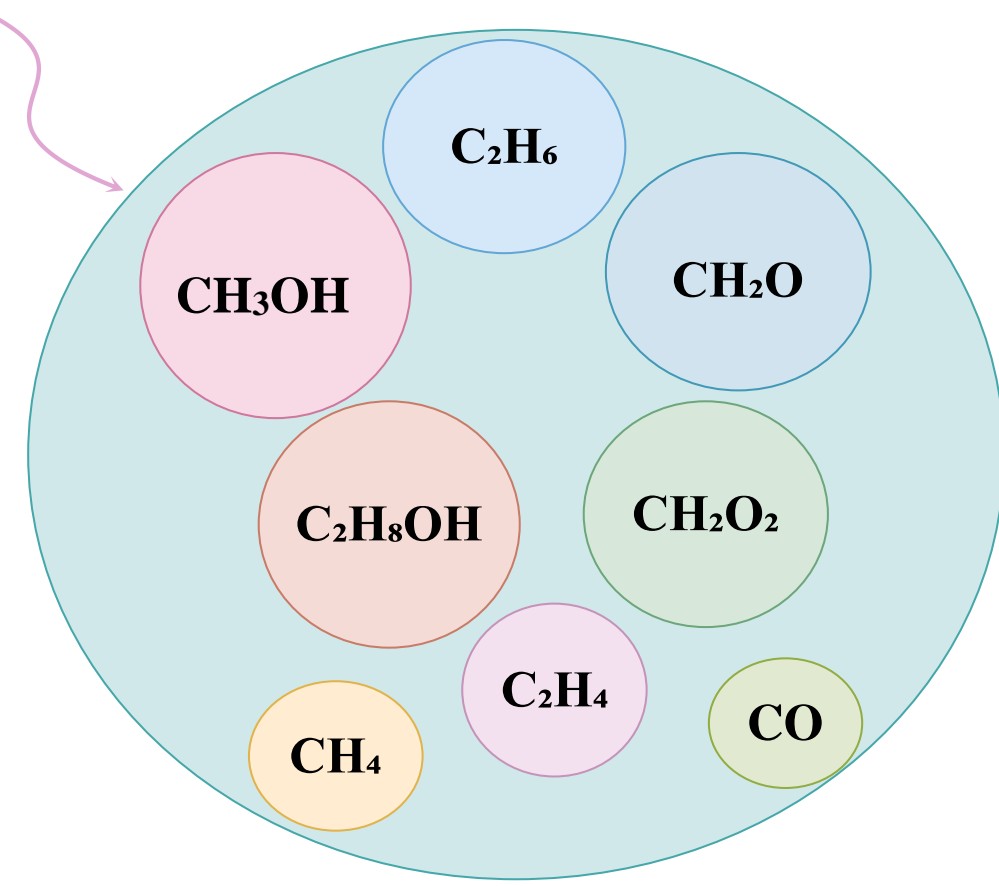

**Figure 2.** Different reaction products from $CO_2$ hydrogenation [15].

### 4. $CO_2$ Hydrogenation

The production of hydrogen fuel faces many obstacles, such as in its strategies for storage and transport and its extensive usage. Methane addresses these issues by providing safety for storage and transport in addition to the ability for the fuel to be liquefied as a natural gas and utilized in industrial energy products. The ability to reuse carbon dioxide and eliminate its harm and the vital importance of the additional methane in the industrial sector enlarge the feasibility of exploring this reaction [18]. Natural photosynthesis involves the transformation of carbon-hydrogenation into fossil fuels. As per this concept, $CO_2$ hydrogenation could be the optimum method for recycling the emissions produced from burning fuel. Figure 3 presents different stages of the methanation process. $CO_2$ is known as a thermodynamically stable compound that is unreactive under ambient conditions. However, significant improvements have been achieved in the transformation of $CO_2$ into a variety of products, such as formic acid, carbon monoxide (CO), methane, and methanol via direct or thermal hydrogen reductions [3]. The production rate of methane is affected by factors such as the reaction temperature, pressure, and presence of catalyst promoters [10].

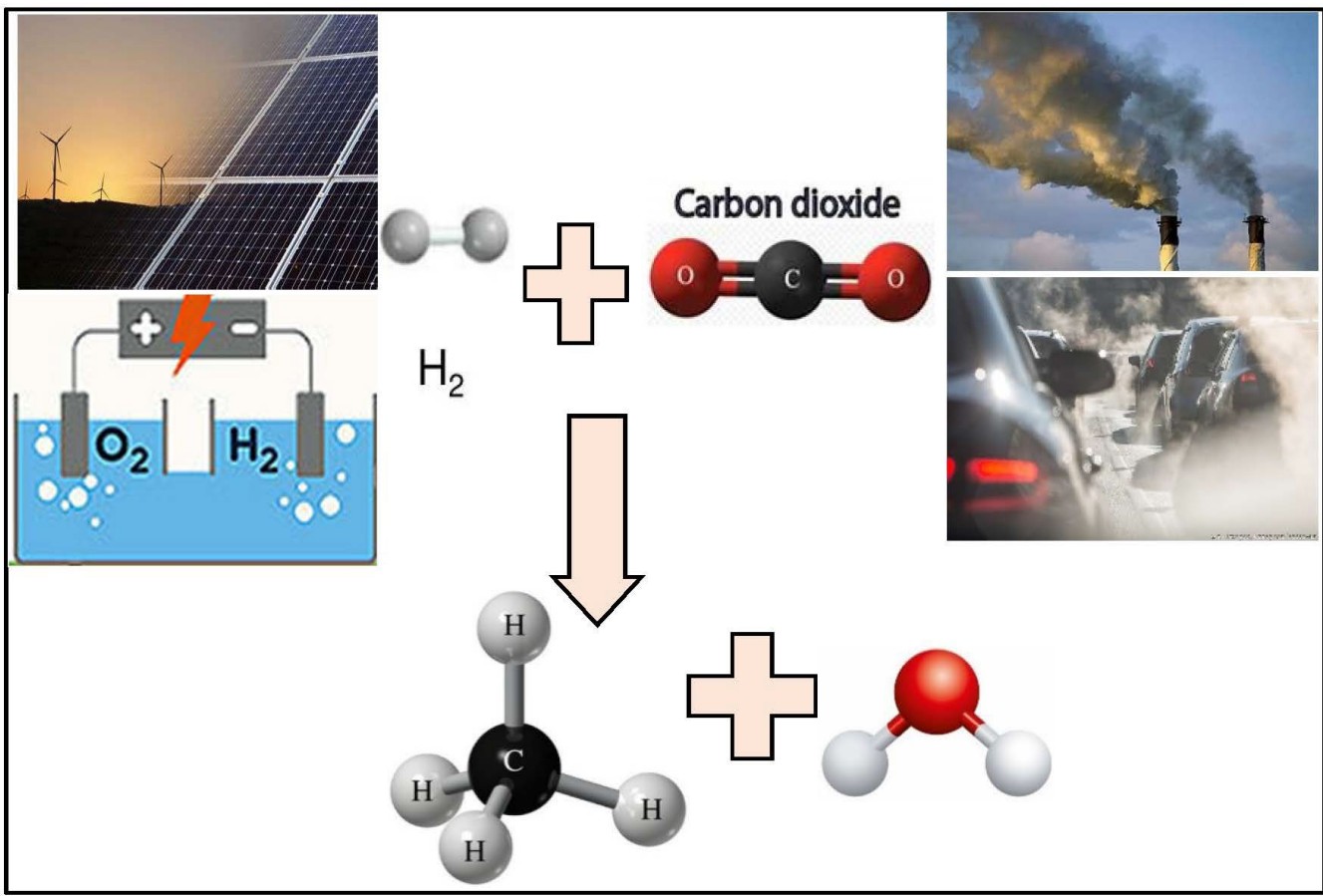

**Figure 3.** The $CO_2$ methanation reaction process.

*Thermodynamic Efficiency*

Thermodynamic studies have indicated the superior conditions of the Sabatier reaction. These studies have provided details about suitable reaction conditions such as temperature, pressure, and reactant ($CO_2$ and $H_2$) concentrations. At low temperatures, $CO_2$ conversion into $CH_4$ produces a high yield. According to the thermodynamic equilibrium, optimal operation conditions ensure the direction and reactivity of the reaction. By raising the temperature, the process of the reaction could go in reverse way toward the methane steam reformation reaction. Increasing the amount of $H_2$ and $CO_2$ gas in the reaction boosts $CH_4$ formation [19]. It is well known that hydrogenation of the $CO_2$ process suffers from many limitations, as shown in Figure 4. The major issues are related to economics and maintaining suitable reaction conditions to obtain stable process efficiency. The hydrogen gas produced from the splitting of water via photo or electrolysis processes becomes sustainable, as conventional $H_2$ production processes are energy consuming. The utilization of solar and wind energy to drive $H_2$ production lowers the cost of the overall $CO_2$ methanation process. Increasing reaction temperatures cause catalyst sintering, which leads to a minimization of the activity and the lifetime of the catalysts. Figure 5 shows the effect of high temperatures on catalysts, where Ni particles are agglomerated and enlarged in size. Additionally, the active sites on the catalyst surface are blocked by deposited species, such as coke. On the other hand, decreasing the reaction temperature results in insufficient thermal energy needed to initiate the $CO_2$ hydrogenation reaction; hence, the catalyst activity is reduced significantly. To tackle this issue, the operating reaction conditions for the process should be optimized [12].

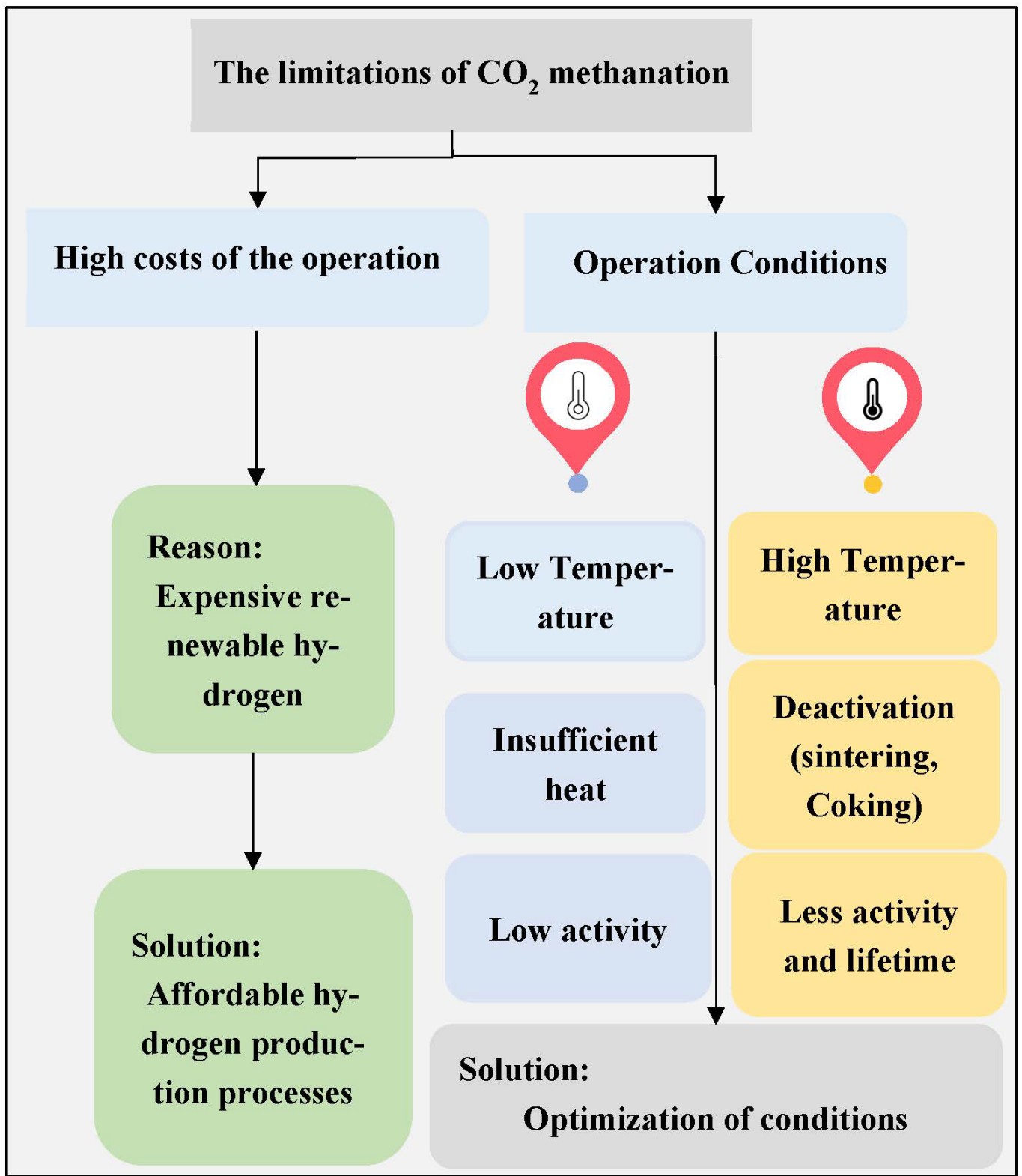

**Figure 4.** Different challenges involved in $CO_2$ methanation [12].

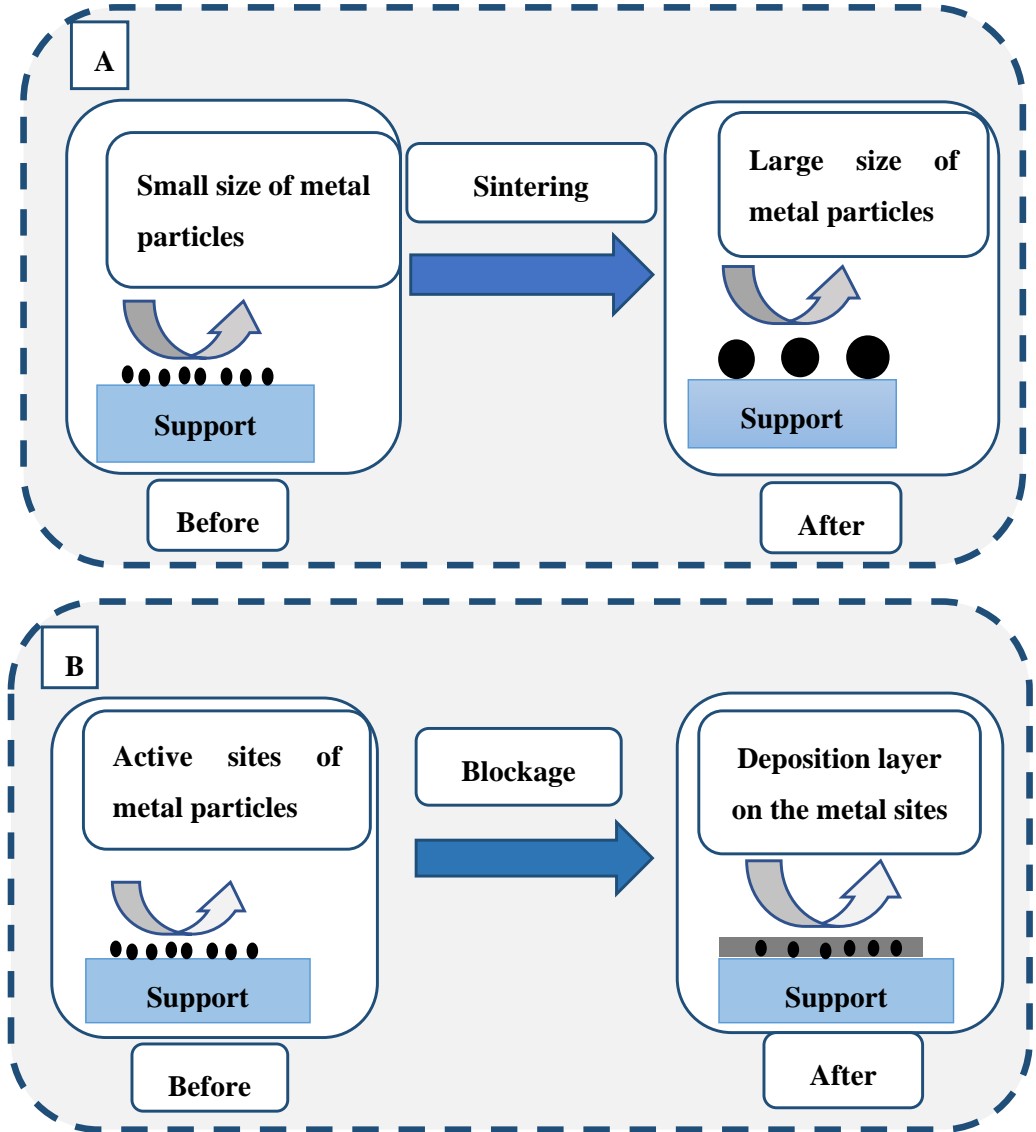

**Figure 5.** Deactivation of Ni-SiO$_2$ catalysts; (**A**) The sintering and (**B**) The layer deposition.

## 5. Catalysis

To achieve effective and efficient CO$_2$ conversion and productivity in valuable chemicals, the CO$_2$ conversion process should be improved using catalysts with high activity, selectivity, and stability [10]. The critical characteristics for designing the optimal catalysts is provided in Figure 6. The development of catalysts with higher intrinsic reaction rates and TOFs should be continued. Furthermore, the catalysts should be stable for a long time in terms of CO$_2$ conversion and CH$_4$ selectivity. However, to date, there are no general guidelines and no role models by which new improvements can be made in preparing catalysts. In this respect, to prepare catalysts with higher activity and to simultaneously prepare the guidelines to synthesize them, researchers have tested many combinations of metals and supports and evaluated their performances in terms CO$_2$ conversion, CH$_4$ selectivity, and catalytic stability. In this review, we have extracted important information regarding the choice of metal, the importance of SiO$_2$ as support, the desirable size of metal nanoparticles, the dimensionality of the metal nanoparticles, and SiO$_2$ support. Furthermore, electronic interactions between a solid support and metal particles very sensitively affect catalytic activity and product selectivity.

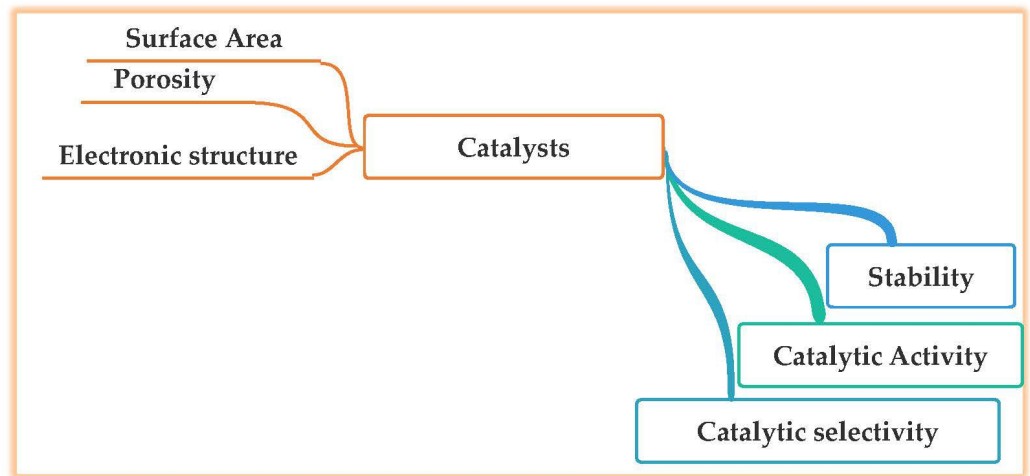

**Figure 6.** Criteria for efficient stable methanation catalyst [14].

$CO_2$ hydrogenation has been performed using catalysts with a variety of metals and supports. The study stated that noble metals such as Ru, Rh, Pt, Pd and transition metals such as Fe, Co, and Ni have the capabilities for $CO_2$ conversion to $CH_4$. Supported materials include metal oxides and basic oxides, such as $Al_2O_3$, $TiO_2$, $SiO_2$, $ZrO_2$, and $CeO_2$, which is massively investigated for $CO_2$ methanation. Recently, research teams have tested the utilization of structured catalysts, such as metal organic frameworks (MOFs), nanosheets, foams, and fibrous materials. The metal used is the main criteria for determining the activity and selectivity of the prepared catalysts for the Sabatier reaction. The catalysts supported on carbide, nitride, and sulfide exhibited sufficient activity and selectivity for $CO_2$ methanation [12]. Different selections were used as catalytic materials for the methanation reaction, and the major advantages and disadvantages of both chosen elements in the catalyst's synthesis are shown in Figure 7.

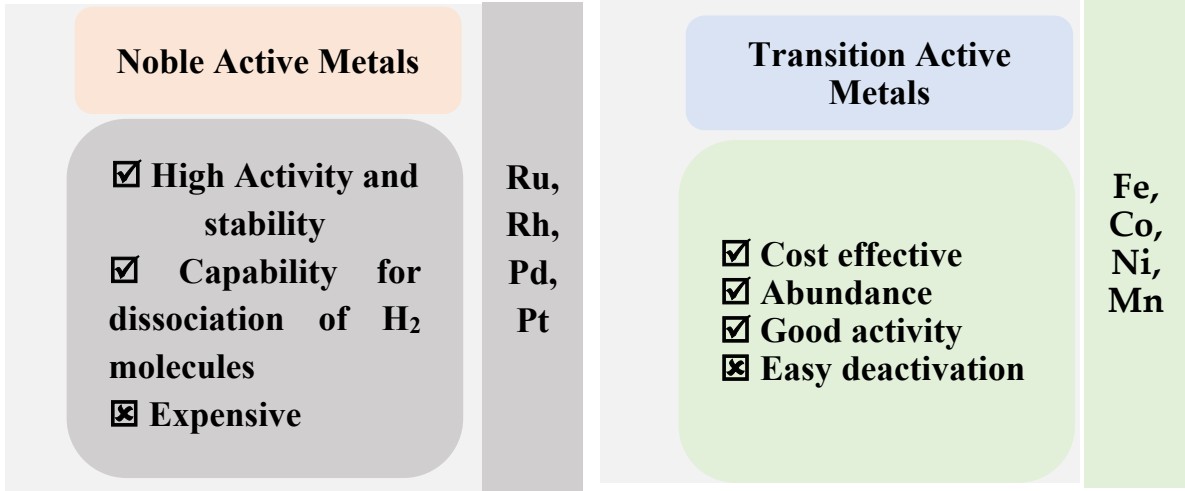

**Figure 7.** The noble metal- and the transition metal-based methanation catalysts [12].

### 6. Noble Metal-Based Catalysts

The growth in the reaction is related to the metal oxidation state as it oxidized by $CO_2$. Noble metal catalysts improve the efficiency of $CO_2$ conversion to $CH_4$, but the high price limits their industrial applications [10]. The noble metals Ru and Rh provide high catalytic activity and stability for $CO_2$ methanation at low temperatures. Due to a lack of abundance and the expense, noble catalysts are unsuitable for large-scale implementation [20].

### 6.1. Rhodium (Rh)

Rhodium is the most studied metal for the $CO_2$ methanation reaction. $TiO_2$ is the most used support for Rh in low temperature hydrogenation reactions. Supported Rh catalysts have been proven to have high activity and chemical stability for $CO_2$ methanation at low temperatures, which is caused by their ability to separate hydrogen molecules from hydrogen atoms [10]. The highest achievements of $Rh/TiO_2$ catalysts are related to their excellent electronic metal–support interaction [21]. This eases the breakage of the bond between carbon and oxygen (C=O), resulting in superior catalytic activity. It has been found that loading different amounts of Rh leads to the formation of different metal particle sizes. The large particles have higher activity compared to the smaller sizes in low-temperature reactions ranging from 130 to 150 °C [10].

### 6.2. Ruthenium (Ru)

Ru has high catalytic activity in the methanation reaction. The support used, promoters, and metal dispersion affect the catalytic activity and selectivity [10]. The ability of supported rhodium catalysts to separate the two hydrogen atoms in a hydrogen molecule causes its distinguished catalytic activity and chemical stability for $CO_2$ hydrogenation at low temperatures [22]. Ru showed the highest catalytic activity, stability, and selectivity. The supported material, with the ability to be reduced as in ($CeO_2$, $TiO_2$, $ZrO_2$), exhibited significant activity with Ru compared to with other supports, such as ($Al_2O_3$, $SiO_2$). The Ru being supported on $CeO_2$ enhanced the reducibility and the availability of oxygen vacancies in comparison to the unpromoted $CeO_2$ [23]. It was found that using Ru with Ni in bimetallic catalysts highly increased the activity and stability. Ru and Rh inhibited deactivation by preventing the sintering and deposition of the carbon [24].

### 6.3. Palladium (Pd)

Pd-based catalysts have been proven to enhance catalytic activity through hydrogen dissociation, where they generate hydrogen atoms to facilitate the methanation reaction. These free hydrogen atoms bond to the carbonate species on the surface [10]. At low temperatures, the supported Pd catalysts are selective for $CO_2$ conversion to methane, while at high temperatures, the catalysts produce lower $CH_4$ compared to the nickel catalyst. This is due to the good water–gas shift catalyst of the Pd with a small formation of CO. Additionally, the $Pd/SiO_2$ catalysts with modifications formed a small amount of unwanted CO at high temperatures of 450 °C in the methanation reaction [25]. Metal oxides prevent the desorption of CO, which reduces the formation of CO. At 450 °C, the sites of Pd and Mg achieve a high methane selectivity of 95% and methane production of 59% [10].

## 7. Transition Metal-Based Catalysts

Transition metals such as iron, copper, cobalt, and nickel increase catalyst performance in the activation and reduction processes of $CO_2$ hydrogenation [10]. Transition metals (Fe, Co, and Cu) have promoted the performance of Ni-based catalysts. Their 3d shells increased the activity and anti-sintering properties during the reaction [7]. Kang et al. investigated the loading effect of Fe on a $NiAl_2O_3$ catalyst for methane production from $CO_2$ conversion and found that the superior performance was gained by the $Ni_{0.7}Fe_{0.3}/Al_2O_3$ catalyst. However, the high Fe loading amount resulted in the water–gas shift reaction. Due to this, optimizing the metals ratio of Ni and Fe formed a high quantity of methane [26]. Promoting the supported material with CO provided high activity and stability [27]. The strategy of Fe and Co bimetallic promotion of the Ni-based catalyst for $CO_2$ methanation produced significant $CH_4$ compared to the monometallic technique [28,29]. Mn was used with $Ni/Al_2O_3$ to improve the methanation of CO and $CO_2$ due to its ability to widely disperse Ni. Burger et al. prepared the $NiMn/Al_2O_3$ catalysts using the co-precipitation method, which provided high medium basic sites, a high $CO_2$ adsorption capacity, and

high catalytic activity [30]. The addition of Fe provided a sufficient increase in the $Ni/ZrO_2$ catalyst performance in $CO_2$ hydrogenation at low temperatures in comparison to that of other additives such as Co and Cu. Fe enhances Ni dispersion and reduction and the partially reduced zirconia, which eases the adsorption and dissociation of molecules of $H_2$ and $CO_2$ [31].

## 8. Nickel-Based Catalysts

Nickel-based catalysts exhibit high performance in carbon dioxide hydrogenation with lower costs. Affective methanation catalysts have been detected to be Ni from the transition metals and Ru and Rh from the noble metals. Ni is commonly used in research for its affordable costs and high availability. The disadvantages of using Ni-based catalysts are the high temperature activity of the nickel, its lack of dispersion, and the reducibility and sintering of nanoparticles. These issues can be solved via the cooperation effect from the addition of a transition or noble metal that is bimetallic. The metal oxide shows a positive impact on the behavior of Ni-based catalysts [32]. To ensure sintering resistance and the active nickel-based catalysts at low temperatures, the structure, dispersion, and interaction between the metal and support should be taken into consideration during design preparation [33].

To improve the performance of Ni-based catalysts, the supported material and promoters need to be modified to optimize the $CO_2$ conversion and methane selectivity. Material with large surface areas, such as $SiO_2$, $Al_2O_3$ and $ZrO_2$, are commonly applied as support for Ni-based catalysts due to the significant abilities of Ni to be dispersed and achieve methane production [34]. Scientists have studied nickel catalytic activity using a variety of supported materials, such as $Al_2O_3$, zeolites, $SiO_2$, $CeO_2$, $ZrO_2$, and $Ce–ZrO_2$, and, more recently, explored the possibility of using hydrotalcite, carbon nanotubes, and W–Mg oxides, shown in Figure 8. Each one of these supports has strengths and weaknesses; for instance, alumina-supported nickel is well known as a high achiever, but it suffers from instability in the elevated temperature of the reaction. Additionally, the impact of the addition of a promoter has been examined [10]. Each type of support material provides the basic improved features of nickel-based catalysts as shown in Figure 9.

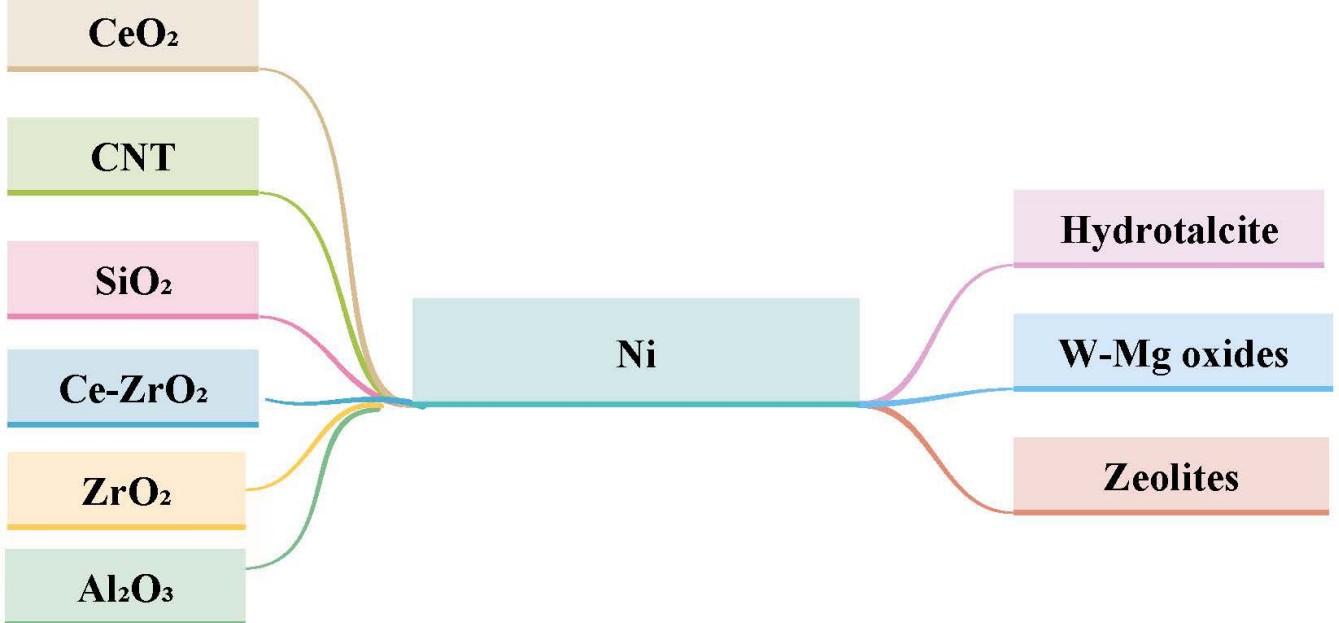

**Figure 8.** Illustration of the different supported materials for Ni-based catalysts [10].

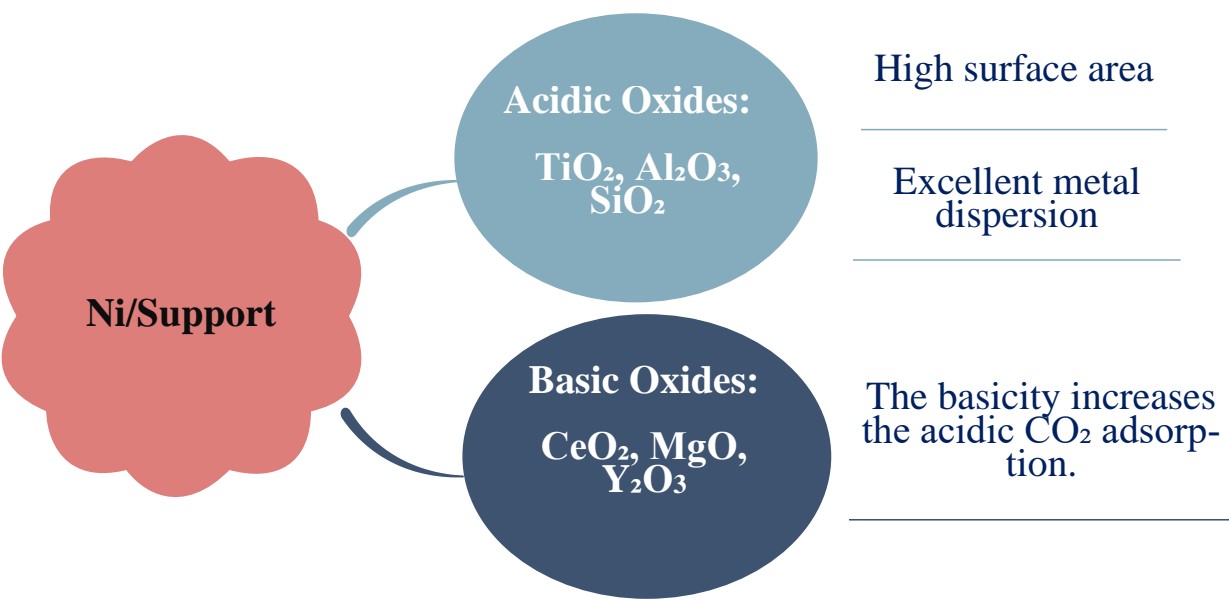

**Figure 9.** Distinct types of support materials for Ni metal [12].

The supported material determines the form of the active sites, the adsorption, and the catalytic activity of the catalysts. This enlarges research into the synthesis of the high metal dispersion support. Vance and Bartholomew prepared a nickel catalyst with different support material to investigate the metal–support effect. The observed catalytic activity and methane selectivity increased in the sequential order of $Ni/TiO_2 > Ni/Al_2O_3 > Ni/SiO_2$ while increasing the electronic interaction between the metal and the support. This is according to the differences in the bonding and the activity of the formed chemisorbed intermediates [35].

*8.1. Nickel Loading Effect*

It has been found that using the proper loading amount of metal improved $CO_2$ methanation, resulting in better dispersion, resistance against metal sintering, and interaction between the metal and support [36]. The amount of metal content on the supported material significantly affected the characteristic and stability of the catalysts. Aziz et al. investigated the nickel content on Ni/MSN catalysts for a methanation reaction. According to the gained data, raising the Ni loading from 1 to 10 wt.% worked effectively with lowering the crystallinity, the surface area, and the basic sites on the surface of the catalysts. The reaction reactivity of the $CO_2$ conversion increased with increasing the Ni loading from 1 to 5% where there were no noticeable enhancements from 5 to 10%. This emphasizes the importance of adjusting the amount of the metal and the presence of basic sites to optimize methanation activity [37]. Liao's research team observed improvements in the catalyst's behavior when accompanied with boosts in the nickel loading [38].

The co-precipitation method was used to form the catalyst $NiCeO_x$ with different amounts of Ni content ranging from 0.3 to 0.8 in CO and $CO_2$ methanation reactions. Increasing the Ni loading led to a large pore volume and pore diameter. The prepared catalyst with the highest Ni content, $Ni_{0.8}Ce_{0.2}O_x$, provided superior performance in the two reactions [39]. The hydrotalcite catalysts were prepared through co-precipitation, where a series of oxide samples were synthesized with a constant Mg\Al ration, and differing molar ratios of nickel ranged from 10.3 to 42.5 wt.% in $CO_2$ hydrogenation. The study indicated that, at low temperatures (250–300 °C), increasing the Ni concentration enhances the activity and selectivity of the reaction due to the lower interaction between the nickel and the supported material [40].

*8.2. The Pretreatment Effect*

8.2.1. Plasma Treatment

Plasma treatment technology is one modification of the preparation method that enhances catalytic activity by providing dispersed metals, strong interactions, and nanosized particle catalysts [18]. Guo et al. reported a high activity of 84.6% in a Ni-La/$\gamma$-Al$_2$O$_3$ catalyst synthesized via the plasma route for a $CO_2$ methanation reaction at 250 °C. The analysis results demonstrated the high surface area and metal dispersion, the small-scaled particles, and the abundance of Ni on the surface of the catalyst due to the plasma treatment [41].

8.2.2. Reduction Pretreatment

Transition metals with 3D unfilled shells, such as Ni and Co, can be integrated with the pore wall of MCM-41 by using reduction as a means for the pretreatment of the nanosized particles previously created, which exhibited outstanding stability during the reaction at high temperatures [42]. Du et al. formed a Ni catalyst with thermal stability and high dispersion through H$_2$ pretreatment. The study determined the effects of treating the samples with pure H$_2$ by varying the temperature on the activity of the catalysts. The reduction treatment performed on the Ni-MCM-41 samples at 973 K for 30 min resulted in significant activity and selectivity, which reduced the nickel to Ni$^0$, attached to the surface with no aggregation [43].

8.2.3. The Calcination

The 50 wt.% Ni-50 wt.% (Zr-Sm oxide) catalyst with calcination at temperatures of 650 or 800 °C performed superior reactivity in the methanation reaction. Zr$^{4+}$ ions of the tetragonal ZrO$_2$ supported the Ni catalysts exchanged by Sm$^{3+}$ ions during the raising of the calcination temperature. The more embodiment in the Sm$^{3+}$ ions led to more activity for methanation by creating oxygen vacancies, which interact highly with the oxygen in $CO_2$ and diminish the strength of the C–O. The calcination at high temperatures caused a shrink in the surface area and Ni dispersion due to growth in the size of the particles. This emphasizes the importance of optimizing the calcination temperature for the catalyst [44].

## 9. Silica (SiO$_2$) as a Support for $CO_2$ Methanation Catalysts

Out of the different silica-supported material, mesostructured silica nanoparticles (MSN) are massively implemented in a variety of applications, such as medical and catalysis. This is due to the attractive characterization of their structure, with a large area and pore volume and sizes ranging from 1.5 to 10 nm. Another widely studied supported material is amorphous pure silica, such as the MCM-41 or SBA-15, with a hexagonal texture or enormous pore and area, respectively [10]. Its large surface area, distinguishing stability, tuned pore diameter, and controlled morphology result in SBA-15 being commonly used to support catalysts. The structured channels of the SBA-15 material dominate the size of the particles and hinder the agglomeration, which causes enhanced stability. The absence of acidity on SBA-15 prohibits carbon deposition when the temperature increases [45]. One study stated that mesostructured silica nanoparticle (MSN) catalysts supported by different metal loadings, such as Rh, Ru, Ni, Fe, Ir, Cu, Zn, V, Cr, Mn, Al, and Zr, synthesize for methanation via the sol–gel and impregnation methods, enhancing the process. The basic site formed depends on the type of metal, where highly active metals are described as the following: Rh/MSN > Ru/MSN > Ni/MSN > Ir/MSN > Fe/MSN > Cu/MSN at 623 K or above. According to the areal basis, Ni/MSN has the highest catalytic activity and Ir/MSN has the lowest [46].

Phyllosilicate is a layered material integrated with metal ions such as Ni, Co, Cu, and Fe, the separation and adsorption applications of which, attributed to catalysis, make it a promising candidate with great adsorption capabilities and facilitated operationalizing [47]. Its mesoporous form enables the development of dispersed active sites, resulting in prominent catalytic activity. Its large surface area, stability, and structured channels provide the

silica-structured pores with enormous dispersion and a strong hold of particles [48]. Shape-controlled Pd nanoparticles are supported on a mesoporous silica shell to form Pd@SiO$_2$ catalysts. A comparison of catalytic efficiency between the CO$_2$ methanation reaction and the impregnated Pd/SiO$_2$ catalyst, which caused deactivation, showed that the latter was more stable, with no particles sintering in the shell catalysts. Catalytic selectivity for CO$_2$ methanation is based on the mean coordination number, which indicates adsorption ability. The highly coordinated Pd (111) facets provide lower activity and CH$_4$ selectivity compared to Pd (100) [49].

## 10. Ni-SiO$_2$ Catalyst

Ni/SiO$_2$ catalysts suffer from a lack of interaction between Ni and the supported silica, which leads to metal sintering at elevated temperatures. This has driven scientists to attempt to enhance the metal–support interaction in Ni/SiO$_2$ catalysts [50]. Ni catalysts have been prepared using two different methods, the sol–gel and the impregnation routes, with mesostructured silica nanoparticles (MSN) and other supports, such as MCM-41, SiO$_2$, $\gamma$-Al$_2$O$_3$, and HY, for CO$_2$ hydrogenation. The catalysts have been estimated and listed according to the highest activity, where MSN comes at the top of the list and AL$_2$O$_3$ comes last (Ni/MSN > Ni/MCM-41 > Ni/HY > Ni/SiO$_2$ > Ni/$\gamma$-Al$_2$O$_3$). The particle porosity structure of the Ni/MSN catalyst promotes catalytic activity by allowing movement of the reactants and the products throughout the reaction. In comparison to Ni/MCM-41, the Ni/MSN's performance is comprehensively high at an elevated temperature rate, as the basic sites enable the wide spread of CO$_2$ in the pores of the catalyst. In terms of the reaction mechanism, carbon species are generated in oxygen vacancies while hydrogen molecules are converted to hydrogen atoms on Ni sites, bonding to the carbon atom and producing synthetic natural gas. The Ni/MSN catalyst has been proven to have excellent stability and a deactivation resistance of up to 200 h [51].

Recent research has explored the use of silica or ordered mesoporous silica (OMS) in supporting Ni catalysts for CO$_2$ methanation, which shows promising possibilities. Pure OMS supports different types of catalysts, such as Ni/SiO$_2$, Ni@SiO$_2$ core-shell, and Ni-SiO$_2$ microspheres. The supported silica suffers from acidity, which limits the CO$_2$ interaction. This can be solved through the addition of the basic oxides, which reduces the acid level on the surface and enhances the CO$_2$ interaction. Therefore, the ordered mesoporous form enriches the catalysts with massive Ni dispersion [8]. Different silica models can be used to form Ni-phyllosilicate, but the preparation of ordered mesoporous silica (OMS) is costly and difficult, which has driven scientists to look for a replacement. Absolute silica is gained from rice husks and is consumed on supporting Ni catalysts, such as the Ni-Ru/SiO$_2$ catalyst. Rice husks contain a huge amount of silicon, which has led to the spread of its usage to a variety of applications, such as in electronics, compositions, and adsorbents. Utilizing the extracted silica from rice husk ash to fabricate Si-MCM-41 is considered an environmentally friendly and cost-effective process. For these reasons, research on using rice husks as source for synthesizing Ni-phyllosilicate catalysts has increased [50].

### 10.1. Different Preparation Methods for Ni/SiO$_2$ Catalysts

Catalyst preparation is a major important step in the manufacturing of catalysts. This is because the catalyst preparation stage is complex and consists of many details that must be known and clear to the catalyst manufacture. There are a large number of details in the preparation stage, which undoubtedly affect the final properties of the catalyst, especially the selectivity and catalytic efficiency of the prepared catalyst. The various techniques for synthesizing supported nickel catalysts are shown in Figure 10. In this part, we will discuss the most important preparation methods for synthesizing Ni/SiO$_2$ catalysts and the characteristics and problems of these methods.

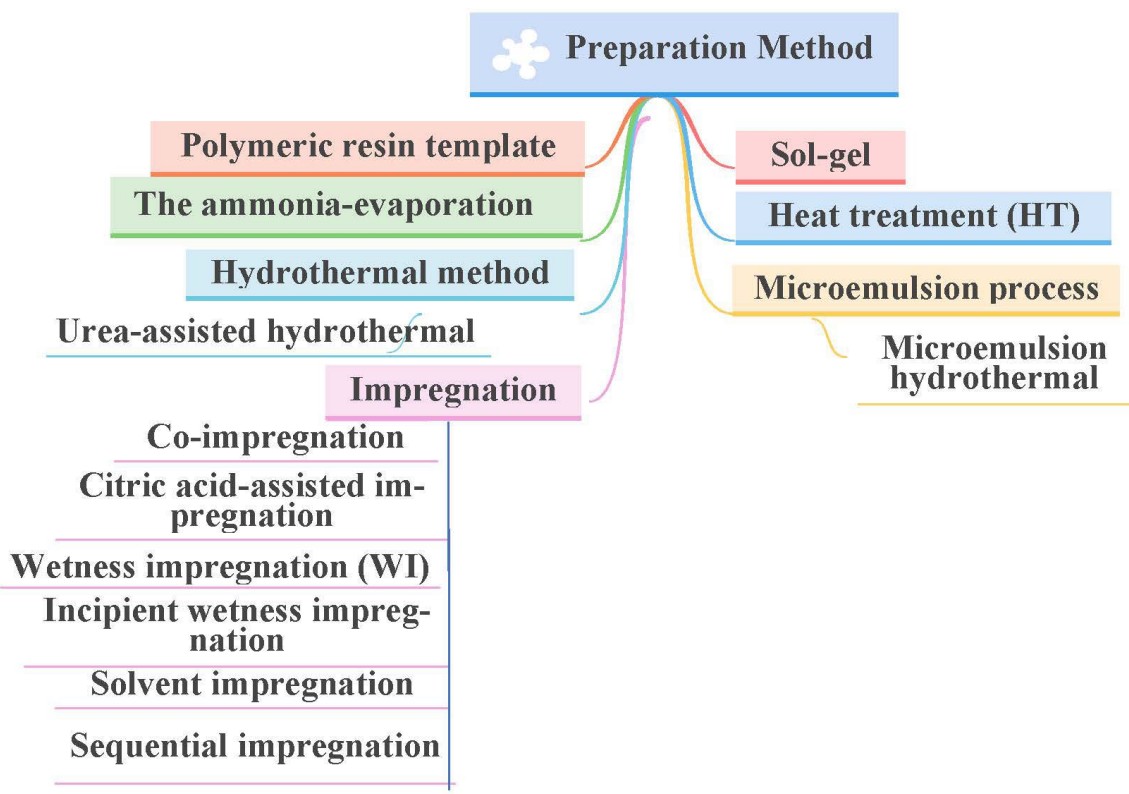

**Figure 10.** Different synthesis strategies for preparation of Ni-SiO$_2$-based catalysts.

*10.2. Effect of Preparation Method*

The utilization of the impregnation strategy to form unalloyed Ni/SiO$_2$ catalysts has lowered their catalytic activity and thermal stability for methane production, which emphasizes the importance of the customized frame. The ammonia-evaporation route for preparing Ni/SiO$_2$ catalysts has increased the efficiency and stability of the reaction while the methane selectivity reached 95% below 100% [52]. The ammonia-evaporation method (AEM) has been used to promote the stability of prepared Ni/SiO$_2$ catalysts for CO$_2$ methanation. This method led to a wide nickel dispersion with small-sized particles, down to 4.2 nm, and the layered structure of the silica support offering a large surface area of around 446.3 m$^2$/g, which caused a strong interaction. As a result, the Ni/SiO$_2$-AEM catalyst exhibited an improved yield and outstanding stability for up to 100 h at 370 °C compared to the catalyst formed via the impregnation route [53].

Hongmanorom et al. studied the impact of the synthesis method on catalytic performance. The comparison was between the wetness impregnation (WI) and the ammonia-evaporation (AE) method. Ni and Ni-Mg phyllosilicates prepared via the AE method were highly affected compared to the ones prepared via the WI method. Additionally, the impact of adopting Mg was examined, where the Ni-5Mg/SBA-15 AE catalyst exhibited a high CO$_2$ conversion rate at lower temperatures due to the high number of basic sites and high Ni dispersion. The structure of the phyllosilicate catalysts confirmed that the interaction between the metal and the support resulted in the avoidance of metal sintering [54]. Xu et al. reported a new preparation method for Ni-based catalysts, called the combustion impregnation method, which is a combined version of two methods, the solution combustion synthesis (SCS) method and the incipient wetness impregnation method. A Ni/SiO$_2$ catalyst with the fuel glycine achieved 94.1% selectivity for CH$_4$ production and 66.9% of CO$_2$ conversion at 350 °C, both of which are significantly higher compared to those of a Ni/SiO$_2$ catalyst synthesized using the conventional impregnation method [55].

A nickel phyllosilicate (Ni$_3$Si$_2$O$_5$(OH)$_4$) catalyst is the result of 3D SBA-15 reacting with nickel via the hydrothermal method. By altering the Ni amount in the range of 24.22

to 30.72 wt.%, its large surface area enables the Ni to disperse highly and form small-scale particles from a reaction with heating at 750 °C. This has been compared to the N/S catalyst with the same Ni content but prepared via the conventional incipient wetness impregnation method. The unique characteristics of 3D nickel phyllosilicate results in a small amount of Ni, superior $H_2$ and $CO_2$ adsorption abilities, and a lower activation energy [47]. The conventional $Ni/SiO_2$ has been used to prepare the structured embedment Ni@HZSM-5 catalyst using the hydrothermal method, which reached a 66.2% activity and a 99.8% selectivity at 673 K. The Ni@HZSM-5 exhibited superior activity and stability in the $CO_2$ methanation reaction for 40 h, without noticeable changes in its structure or nickel content, in comparison to the $Ni/SiO_2$ and Ni/HZSM-5 catalysts via the impregnation method. The metal phase of the Ni@HZSM-5 catalyst provided the zeolite with electrons, which raised the value of the BE and resulted in the avoidance of metal sintering [56].

Wang et al. compared two fabrication strategies, wet impregnation and the citrate complex method, and their findings indicated the influence of La addition on Ni/SBA-15 catalysts at low temperatures. The citrate complex method enables a wide Ni dispersion on the surface of the SBA-15, causing improved $CO_2$ adsorption and activation and overall $CO_2$ redaction via $Ni-La_2O_3/SBA-15$ [48]. The SBA-15 formations gained from the two different preparation methods (classical and microwave-assisted) have the same structure, which slightly differs from the addition of Ni and Ni-Ce. Due to the large pores of SBA-15, Ni particles are dispersed on the outer layer of the catalysts, whereas MCM-41, with small pores, contains Ni particles on the external layer that form huge particles. The MCM-41 Ni catalysts exhibit superior performance in anchoring the metal sintering and developing the metal and support interaction. Additive Ce facilitates $CO_2$ activation, resulting in high catalytic activity at low temperatures [57].

$Ni/SiO_2$-promoted catalysts with varying amounts of Mg that have been synthesized using co-impregnation exhibit superior, lifelong performance compared to the catalysts prepared using the sequential impregnation strategy. These improvements are related to the high adsorption and activation abilities of $CO_2$ and Ni, which limits the oxidation and the deactivation of nickel [58]. Wang et al. studied the influence of nickel particle size on the performance of $Ni/SiO_2$ catalysts in $CO_2$ methanation. The particle sizes varied from 3.5 to 7.5 nm, depending on the preparation method, with a constant Ni amount of 2 wt.%. It was found that the small Ni nanoparticle provides more catalytic activity than the large Ni particle on $SiO_2$ as a support. Lowering the size of the Ni particle forms adsorption and activation sites for $CO_2$, which enables the conversion of $CO_2$ into $CH_4$ without CO product [20].

Xu et al. compared two versions of $Ni/SiO_2$ catalysts, one prepared via the conventional impregnation method and the other prepared via the impregnation method using plasma technology, in terms of their adsorption, reduction, and catalytic activity in $CO_2$ hydrogenation. These dependent factors were positively affected in the case of plasma modifications [18]. Paviotti et al. developed a time- and cost-effective synthesis from wasted rice husk ashes to gain meso cellular silica foam (MCF) using the hydrothermal method via microwave irradiation. For Ni-based catalyst preparation, one-pot and incipient wetness impregnation methods were used to form MCM-41 and MCF using cyclohexane as a swelling agent for $CO_2$ reduction at the temperature range of 200 and 500 °C. The interaction between the metal and the support changed depending on the preparation method [58,59]. The results proved the high activity of the impregnation samples over the one-pot samples. Ni/MCF exhibits outstanding $CO_2$ conversion, stability, coke resistance, and $CH_4$ selectivity at temperatures between 350 and 500 °C due to the Ni distribution and the support structure [60]. Gac et al. discovered a new preparation method to form nickel-based catalysts supported by silica microspheres using polymeric resin (Amberlite XAD7HP) for $CO_2$ methanation. The addition of nickel and silica to the resin beads occurred during different stages of the reaction processes, as shown in Figure 11.

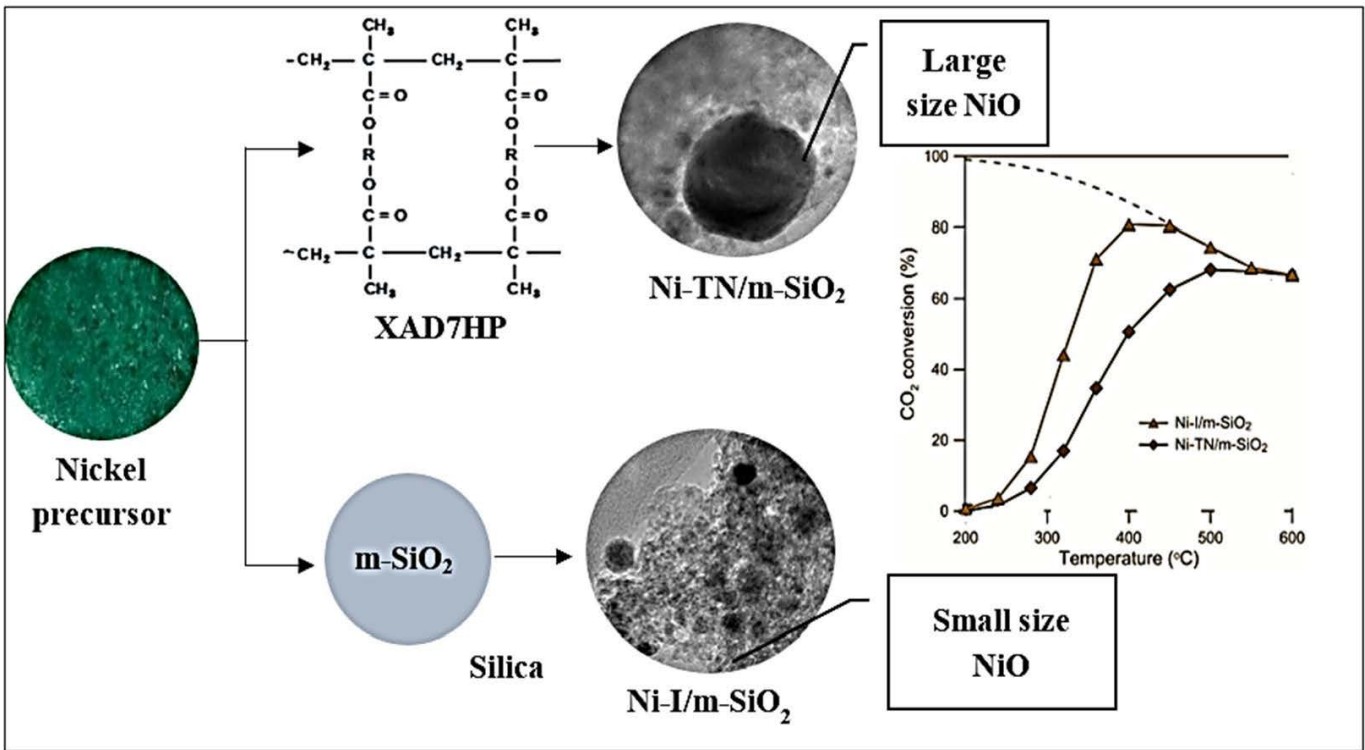

**Figure 11.** Utilization of polymeric resin (Amberlite XAD7HP) in the preparation of nickel-based catalysts for $CO_2$ methanation [59]. (Copyright. Elsevier).

The Ni nanoparticles were generated as small with a hierarchical pore texture when the silica was impregnated prior to removing the template. It was found that the compatibility of the activity and selectivity in the Sabatier reaction corresponded with the broadened surface area and smaller size of the Ni particles. The selectivity and the rate of the reaction showed a distinctive rise as the particle size went down and the active surface sites increased at the temperature range of 220 to 350 °C [59]. Chen et al. revealed the pivotal effect of the fabrication strategy on the nature of the structure, the performance, and the stability of catalysts through the Sabatier reaction by using two different methods, as demonstrated in Figure 12. They used the conventional and the urea-assisted hydrothermal routes to form a Ni-phyllosilicate catalyst from nickel nitrate and silica from a rice husk. The Ni-phyllosilicate was prepared hydrothermally at 180 and 220 °C for the same time duration of 2 days, resulting in about 4.2 wt.% and 10.2 wt.% of Ni in the $N_{180}$/SR and $N_{220}$/SR catalysts, respectively, due to the lower silanol contents after calcination. By applying the modified method with urea and lowering the nucleation time (24 h), the authors synthesized Ni-phyllosilicate ($N_{180}$/SR-U-24) at 180 °C, which contained 22.6 wt.% of Ni. The decay of the urea throughout the reaction led to the consistency of the Ni $(OH)_2$ and $SiO_2$ leaching enabling the facile synthesis of Ni-phyllosilicate. $N_{180}$/SR-U-24 showed distinguishing activity and stability in the reaction of $CO_2$ hydrogenation because of its elevated Ni amounts and the excellent interaction between Ni and the supported material offered by the Ni-phyllosilicate. This proved the effectiveness of the denoted catalyst preparation method [50].

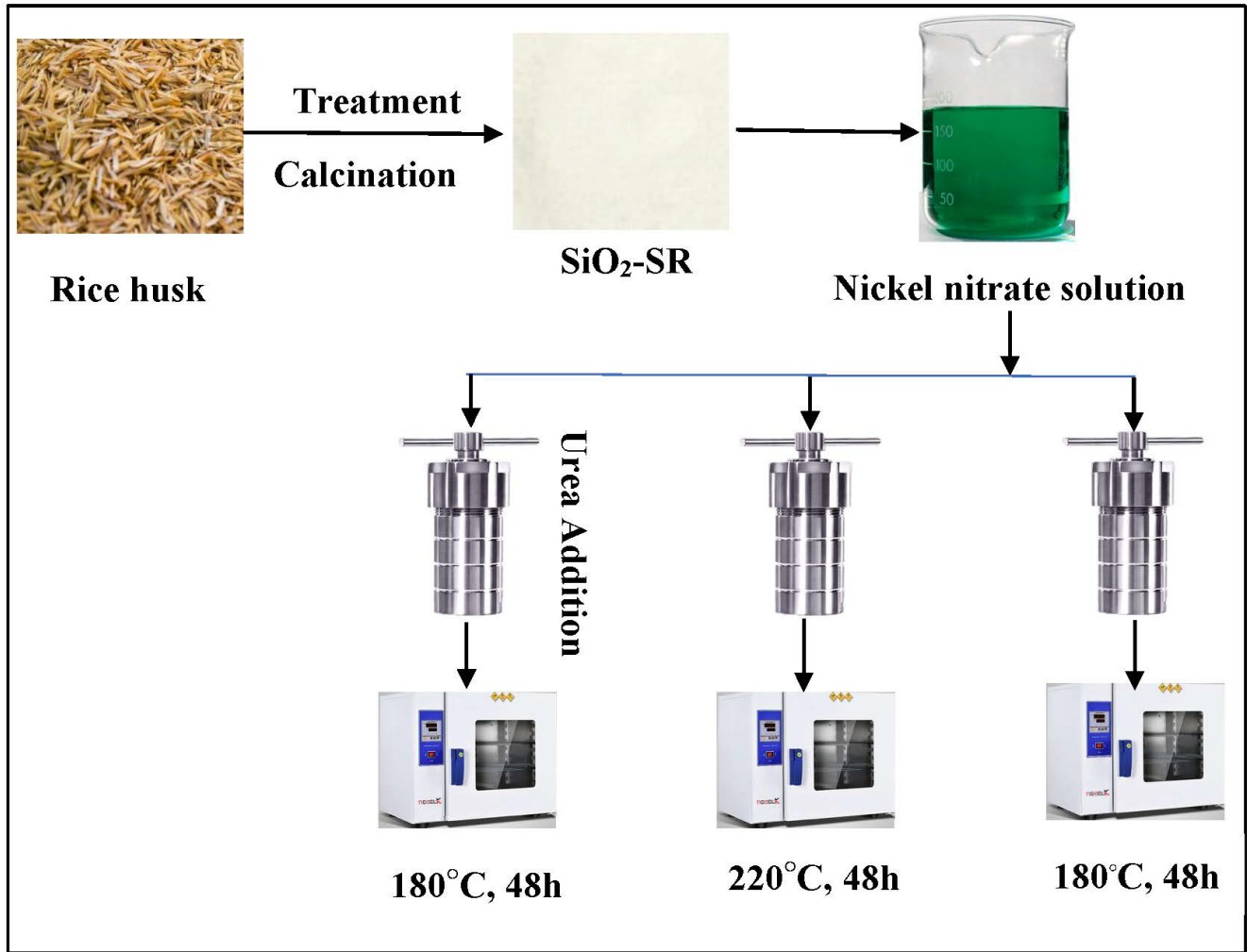

**Figure 12.** Hydrothermal preparation of Ni-phyllosilicate catalyst via the conventional and the urea-assisted method [50].

The metal-organic framework (MOF) has exhibited a remarkable catalytic capability for $CO_2$ methanation. Its expensive preparation method and the lack of stability in its derivatives has driven scientists to find an alternative synthesis route. Ye et al. modified the one-step sol–gel strategy to form a $Ni/SiO_2$ catalyst for the Sabatier reaction at low temperatures with outstanding performance, as illustrated in Figure 13 [52]. Comparing the developed one-pot method to the formal impregnation method, NiO/SBA-15-Op provides a larger surface area, a larger pore volume, and higher Ni dispersion than does the NiO/SBA-15-Im catalyst. Moreover, the broadly dispersed Ni in the channels of the mesostructured NiO/SBA-15-Op catalyst enhances the activity and stability of the reaction to a higher level than does the NiO/SBA-15-Im [45]. The synthesis method of the Ni catalyst affects its metal–support interaction, where the deposition–precipitation (DP) method achieves an important level of interaction in comparison to the wet impregnation (WI) method. Varying the Ni content from 10 to 20 wt.% in two catalysts, Ni/SiC and $Ni/SiO_2$, formed using WI, has been associated with raising the efficiency of the catalytic reaction. In comparison, the activity of a Ni/SiC catalyst with DP preparation has overcome that of the WI preparation sample for the hydrogenation of both CO and $CO_2$. The DP method, used for SiC formation, has allowed for the spread of dispersed Ni and thermal accessibility, which facilitates CO and $CO_2$ conversion reactions [11].

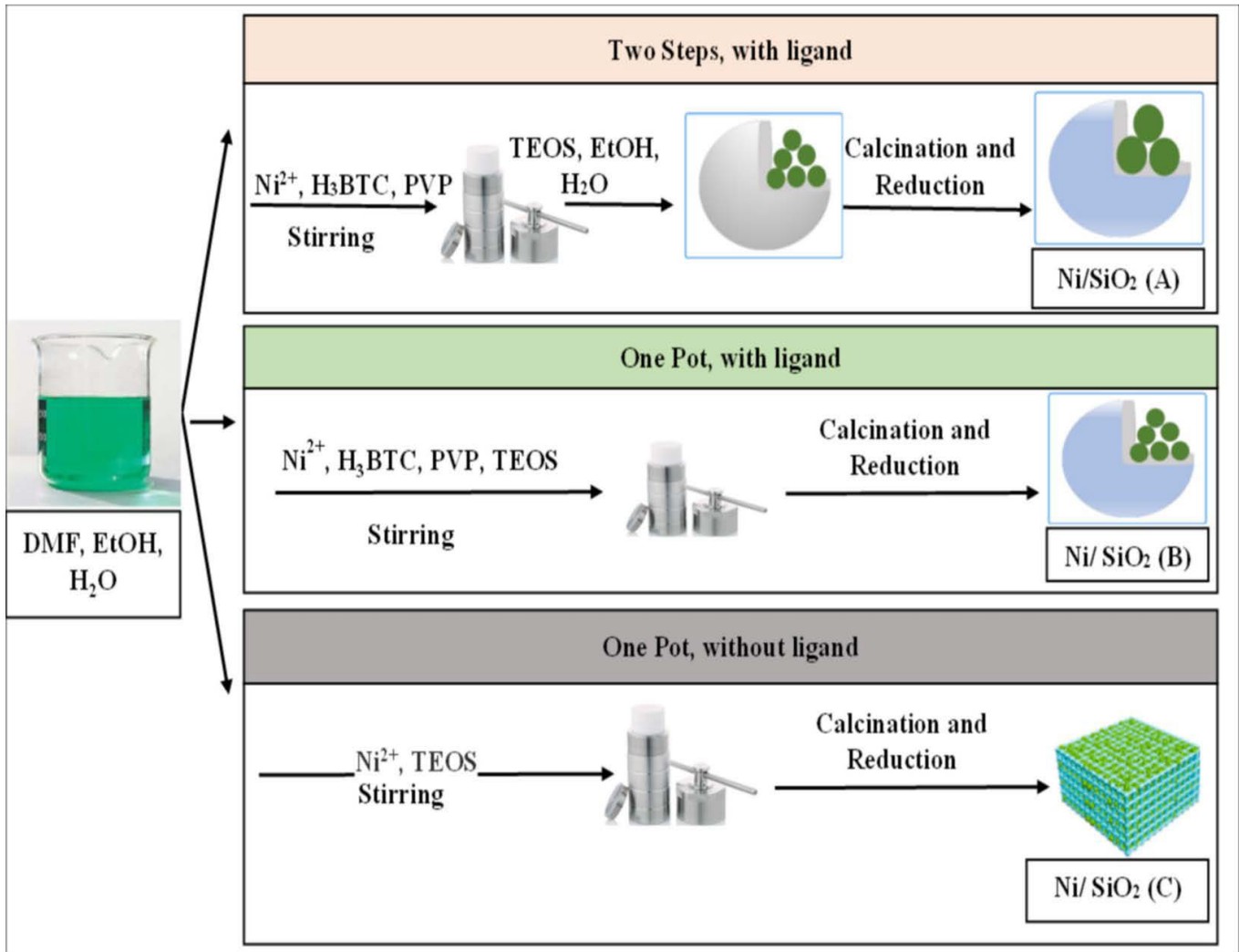

**Figure 13.** Hydrothermal preparation of Ni/SiO$_2$ catalyst [52]. (Copyright. Elsevier).

## 11. Effect of Support Modification

Figure 14 describes support multifunction, which could influence the overall catalytic CO$_2$ methanation activity. Zhu et al. reported the consequences of introducing SiO$_2$, Al$_2$O$_3$, and ZrO$_2$ to a support for the conversion of CO$_2$ to the methane. The additives provided a large surface area and facilitated CO$_2$ dissociation and an excellent interaction between Ni and the support, which separately improved the activity, selectivity, and stability of the reaction based on the nanoparticles applied at temperatures lower than 450 °C [34]. Nickel-based catalysts prepared with different mesoporous structures can support materials such as ZSM-5, SBA-15, MCM-41, Al$_2$O$_3$, and SiO$_2$ through use of the incipient wetness impregnation method for converting CO$_2$ to methane. The catalysts with the highest catalytic activity are ranked in the order of Ni/ZSM-5, Ni/SBA-15, Ni/Al$_2$O$_3$, Ni/SiO$_2$, and Ni/MCM-41. Its basicity and the combined impact of its metal and support cause the highest catalytic activity of the Ni/ZSM-5 catalyst. According to IR spectra, monodentate is a highly reactive molecule compared to the surface formational species in the Ni/ZSM-5 catalyst. Moreover, an Ni/ZSM-5 catalyst has lasted for 100 h without deactivation and exhibited anti-sintering characteristics [61].

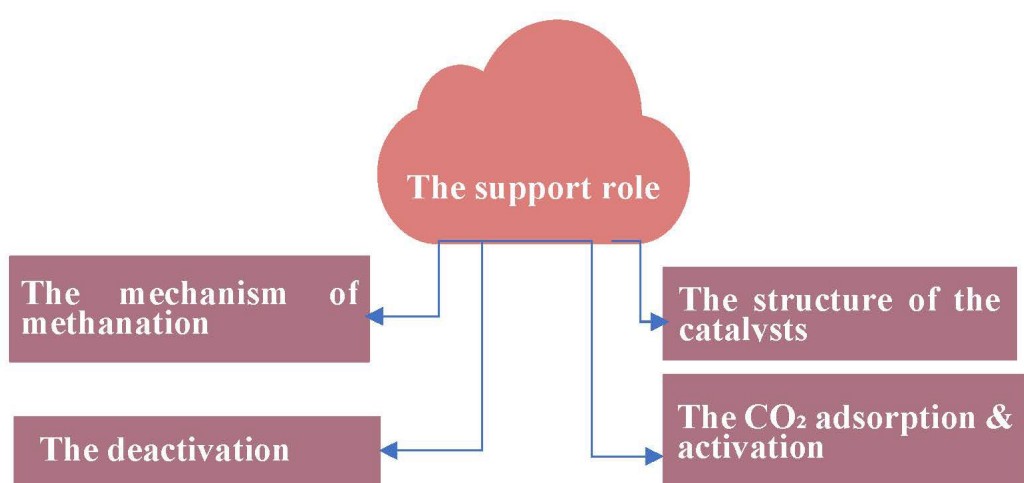

**Figure 14.** Effect of the support on the design of Ni-SiO$_2$-based catalysts [9].

Mihet et al. deposited 10 wt.%. Ni over different oxides (MgO, CeO$_2$, and La$_2$O$_3$) that were mixed with silica support. The presence of these oxides resulted in a significant improvement in Ni dispersion and many active centers activating H$_2$. The Ni/CeSi catalyst exhibited a high basicity and CO$_2$ activation ability in comparison to Ni/MgSi during a methanation reaction at temperatures varying from 30 to 350 °C. In terms of catalytic efficiency, Ni/LaSi proved to be the best catalyst, with 83% CO$_2$ transformation and 98% selectivity at 250 °C. This effectiveness is related to the development of an H$_2$ and CO$_2$ activation property. The examined catalysts kept their pore structures with no changes to either mesopores of small or medium size. No metal sintering or carbon deposition was detected on the catalysts, other than on Ni/CeSi [8].

A microemulsion process has been used to synthesize fibrous silica-mordenite (FS@SiO$_2$-MOR), which achieved 65% CO$_2$ conversion and 68% CH$_4$ selectivity during CO$_2$ hydrogenation with thermal stability for 50 h in comparison to the commercial mordenite (MOR), as depicted in Figure 15. The unique fibrous morphology of FS@SiO$_2$-MOR, with large surfaces, massive oxygen vacancies, and basic sites, allows for CO$_2$ and H$_2$ adsorption. The oxygen vacancies enable the activation of each of the CO$_2$ and H$_2$ molecules in the CO$_2$ methanation reaction [62]. Ma et al. reported a layered catalyst in which Ni foam was covered with a graphene layer and nickel silicate was deposited on the surface, as shown in Figure 16. The presence of graphene oxide between the Ni-SiO$_2$ and Ni foam provided outstanding stability and reactivity for CO$_2$ hydrogenation at the high temperature of 470 °C due to the high interaction between metal and support, significant Ni dispersion, and low activation energy [63].

Zhang and Liu used the citric acid–assisted impregnation method to form a mesostructured cellular foam (MCF) silica as a support for LaNiO$_3$-based catalysts for a CO$_2$ conversion reaction. Calcinating LaNiO$_3$/MCF catalysts at 650 °C provided a strong interaction between and wide dispersion of both the La$_2$O$_3$ and Ni nanoparticles in the pores of the support, as shown in Figure 17. Due to this, the LaNiO$_3$/MCF (30LNOM-C-650) catalyst showed superior catalytic reactivity in comparison to the one (30LNOM-Im-650) synthesized using the co-impregnation method, which reached its highest CO$_2$ conversion and CH$_4$ selectivity at 76% and 97%, respectively, at 450 °C. Additionally, its stability was tested at the same temperature, and it was found to be stable for 100 h with high anti-sintering effects in regard to the high interaction between the metal and the support [64].

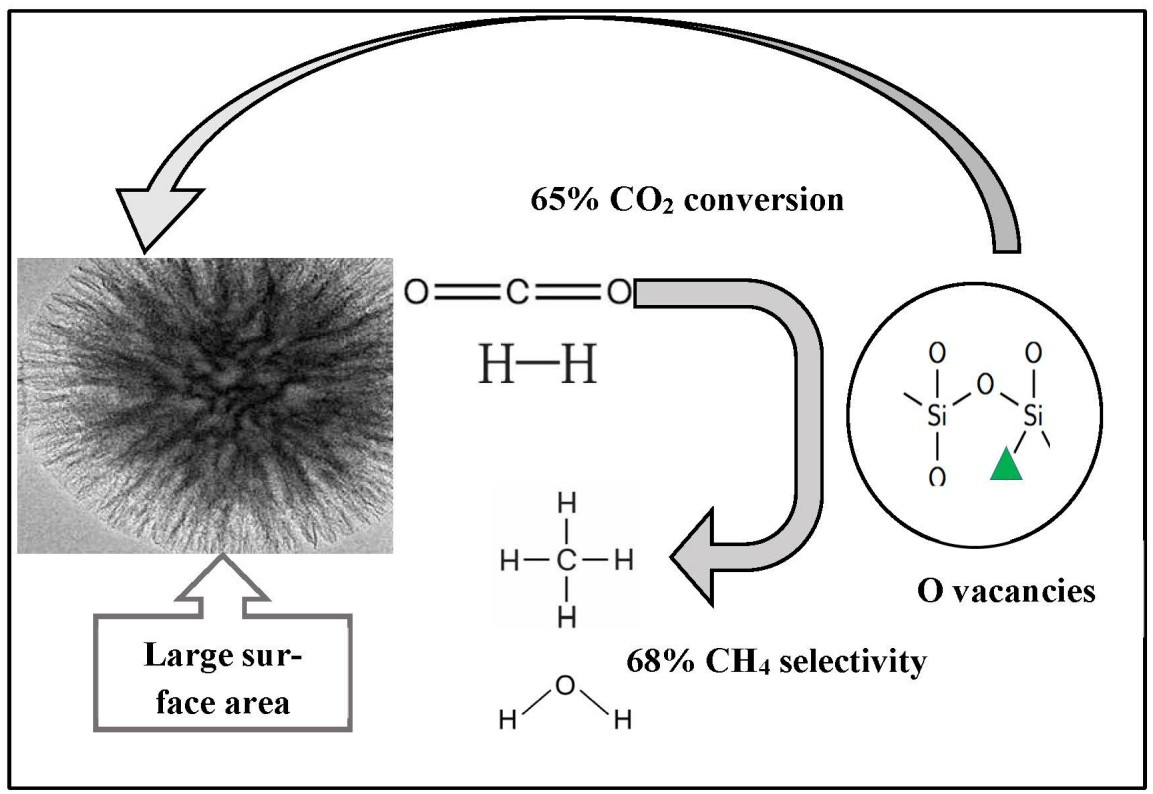

**Figure 15.** FS@SiO$_2$-MOR catalyst for CO$_2$ hydrogenation [62]. (Copyright. Elsevier).

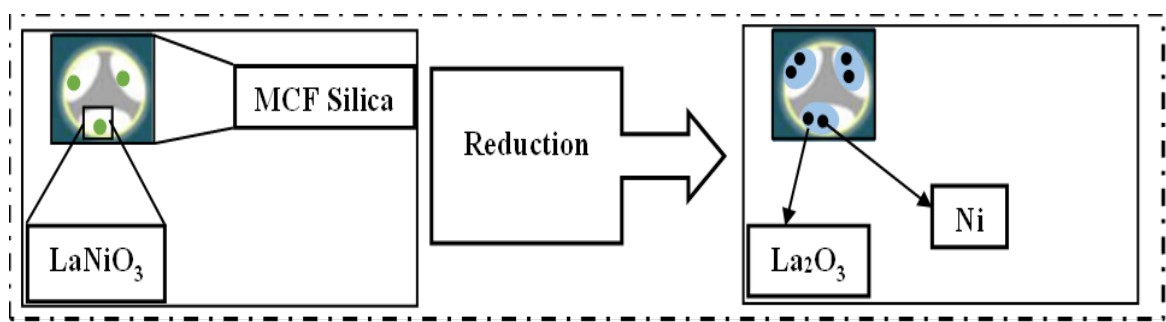

**Figure 16.** Schematic illustration of Ni-SiO$_2$/GO/Ni-foam catalyst [63].

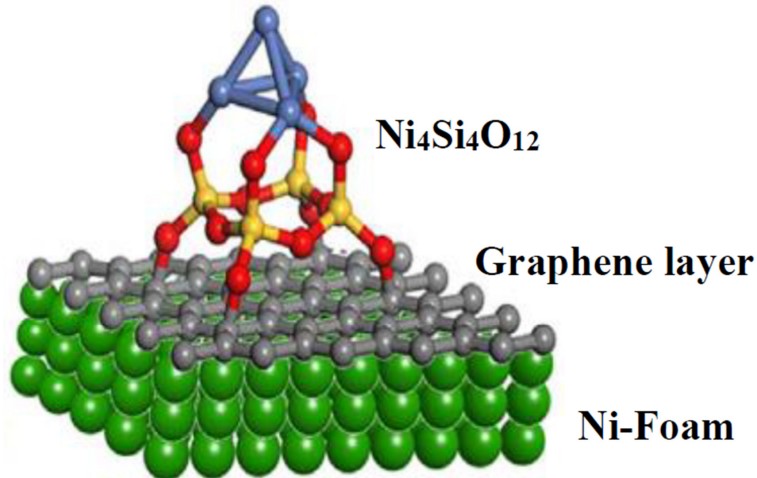

**Figure 17.** Fabrication of the La$_x$NiOM-C-y catalyst [64]. (Copyright. Elsevier).

Lv et al. developed a microemulsion hydrothermal synthesis to prepare Ni catalysts supported by the fibrous KCC-1 nanosphere for $CO_2$ methanation, as shown in Figure 18. The unusual mesoporous structure of the fibrous KCC-1 support materials enlarged the dispersion of the Ni metal by widening the available area. In comparing this to the reference catalysts, the catalyst 20Ni/KCC-1 exhibited a higher activity than did the 20Ni/SiO$_2$ and 20Ni/MCM-41 at low temperatures. The dendrimeric mesoporous structure of 20Ni/KCC-1 led to its noticeable stability for 40 h at a temperature of 400 °C, an impact of the sinter-proof nature of the Ni sites. Kinetically, the high dispersion of Ni on the 20Ni/KCC-1 led to a lower activation energy than that of the reference catalysts [65].

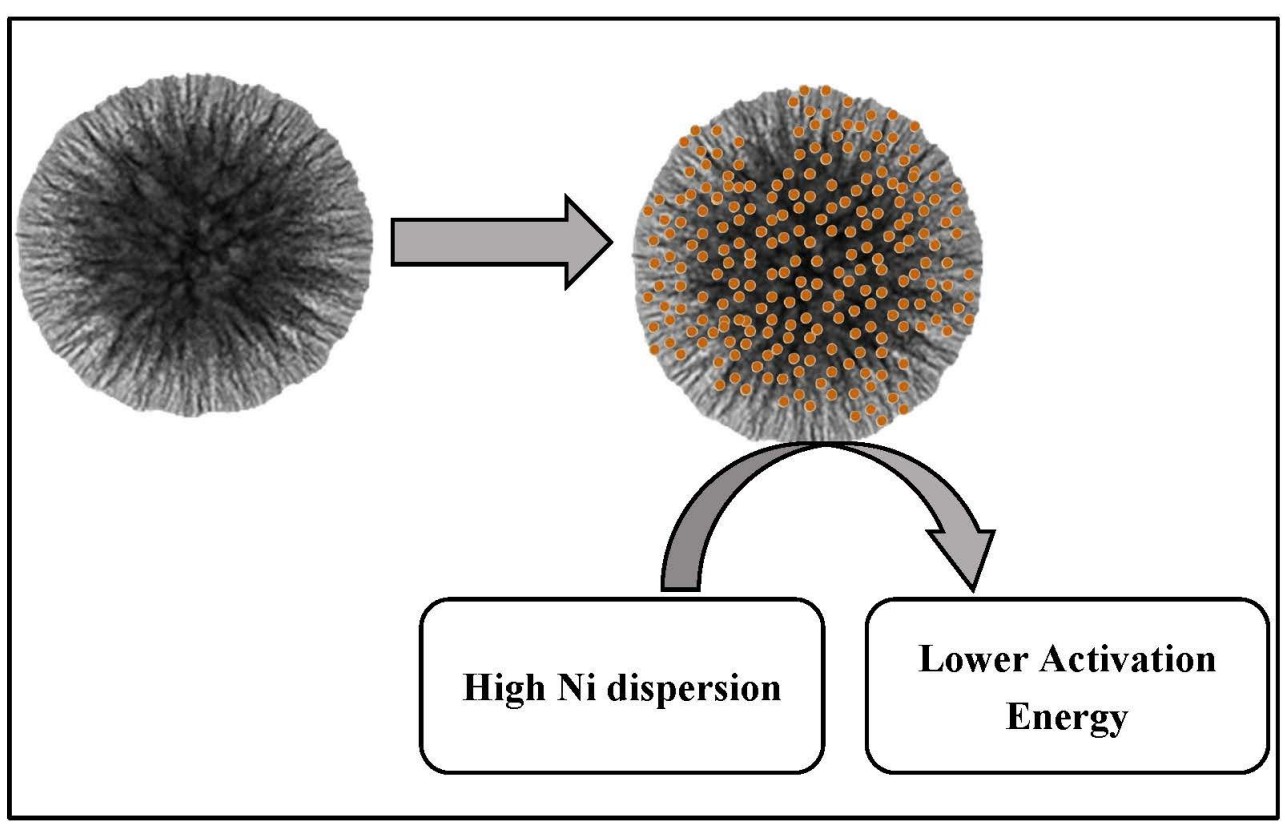

**Figure 18.** Ni/KCC-1 nanospheres for $CO_2$ methanation [65]. (Copyright. Elsevier).

Moghaddam et al. used the sol–gel method to prepare groups of mesoporous alumina-silica Ni-based catalysts with a variety of $SiO_2/Al_2O_3$ ratios employed for the conversion of $CO_2$ to $CH_4$. A high ratio of the Si/Al caused a reduction in the surface area of the supported samples from 254 to 163.3 m$^2$/g and an increase in the Ni particle size from 3.53 to 5.14 nm. A TPR analysis identified the reducible capability of nickel in its conversion from nickel oxide to nickel metal at low temperatures. The authors optimized the catalyst with a 0.5 ratio of $SiO_2/Al_2O_3$, each with the highest stability, deactivation resistance, and catalytic activity, as well as a selectivity of 82.4% and 98.2%, respectively, at 350 °C [66].

Yan et al. pointed out that nickel located on the internal layer of 2D siloxene nanosheets promoted activity, selectivity, and stability in a reaction, which can be gained by adjusting each terminal group of siloxene and the utilized solvent during formation. With the nickel in the center, the rate of $CO_2$ hydrogenation reached 100 mmolg$^{-1}$ h$^{-1}$ with 100% selectivity, as presented in Figure 19 [67]. Moghaddam et al. found that the optimal ratio for an Si/Al catalyst is 0.5, which can be prepared using one of the improved techniques for obtaining high catalytic productivity. Moreover, the optimized Si/Al ratio could be applied to different zeolitic structures.

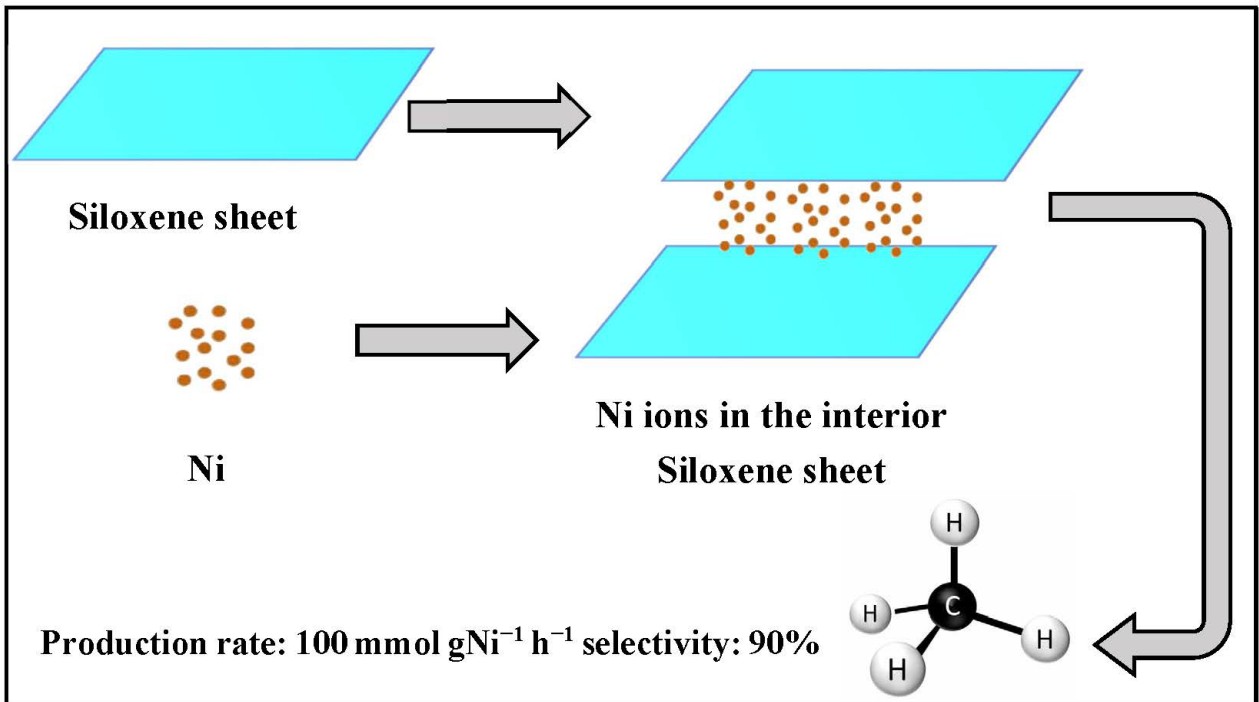

**Figure 19.** Centered Ni on the 2D siloxene nanosheets [67].

## 12. Effect of the Ni Loading

NiO/SBA-15 has shown a high rate of $CO_2$ conversion to $CH_4$ and a selectivity reaching 100% when the loaded NiO amount is increased in two different preparation methods, a simple heat treatment (HT) and the conventional solvent impregnation (SI) method [68]. Liao et al. indicated the effects of Ni loading on both the structure and the catalytic activity of the reaction by using a thermal treatment to prepare a group of nickel silicate catalyst samples for $CO_2$ methanation. According to the study, a rising nickel content was associated with an improvement in activity. The highest achievement was for the NiPS catalyst with a mass of silica equal to 1.6 g and 34.3 wt.% of Ni loading, resulting in more than 80% of $CO_2$ conversion and about 100% of methane selectivity at 330 °C for 48 h. The distinctive nature of the layered silicate, with an abundance of nickel phyllosilicate and the availability of Lewis acid sites, provided high dispersion of Ni and an interaction between the carrier and the active sites [38].

Dias et al. estimated the influence of altering the Ni loading amount in a Ni-Cu/$SiO_2$ catalyst synthesis via wet impregnation for $CO_2$ methanation at varying temperatures from 200 to 400 °C. Increasing the Ni loading was accompanied with raising both the catalytic activity and the methane selectivity. The catalyst with $Ni_{15}$ exhibited a production and selectivity elevated by 55% and 96%, respectively, at a temperature equal to 350 °C. The addition of Cu shifted the selectivity toward CO instead of $CO_2$, which lowered the reaction rate. However, it promoted the catalyst's resistance to deactivation. During the period of the reaction (5 h), the tested samples remained stable at the temperature of 400 °C [69]. Bukhari et al. revealed the effects of differing Ni loadings, between 1 and 10 wt.%, on the SBA-15 fibrous catalysts (F-SBA-15) in $CO_2$ methanation. Methane production increased with increases in the metal from 1 to 5 wt.% and then started to decrease. The improved activity and stability were related to the unique structure of the supported fiber, which provided more Ni dispersion, metal–support interactions, and basic sites. The combination of the texture and the optimized loading results in significant catalyst improvements and can be practiced in many fields [36].

### 13. CO$_2$ Methanation Reaction Mechanism over Ni/SiO$_2$-based Catalysts

The CO$_2$ methanation reaction mechanism is associated with the characteristics of Ni/SiO$_2$ catalysts. Ni/ZSM-5 and Ni/MCM-41 (Ni-supported zeolite) catalysts exhibit the desired high performance. CO$_2$ methanation occurs through two potential mechanisms, (i) formate or carbonate pathway and (ii) carbon monoxide pathway. Methanation reactions over Ni/ZSM-5 catalysts form formate species through the carbonate's hydrogenation rather than the CO path. An IR analysis revealed that the Ni/ZSM-5 catalyst form two types of species, monodentate and bidentate formate, at 1340 and 1600 cm$^{-1}$, respectively (Figure 20). On the other hand, a catalyst of Ni/MCM-41 showed a peak at 1590 cm$^{-1}$ due to the formation of bidentate species. Thus, the high activity of Ni/ZSM-5 catalysts indicates the efficiency of the monodentate path of reaction compared to that of bidentate formate methanation [61].

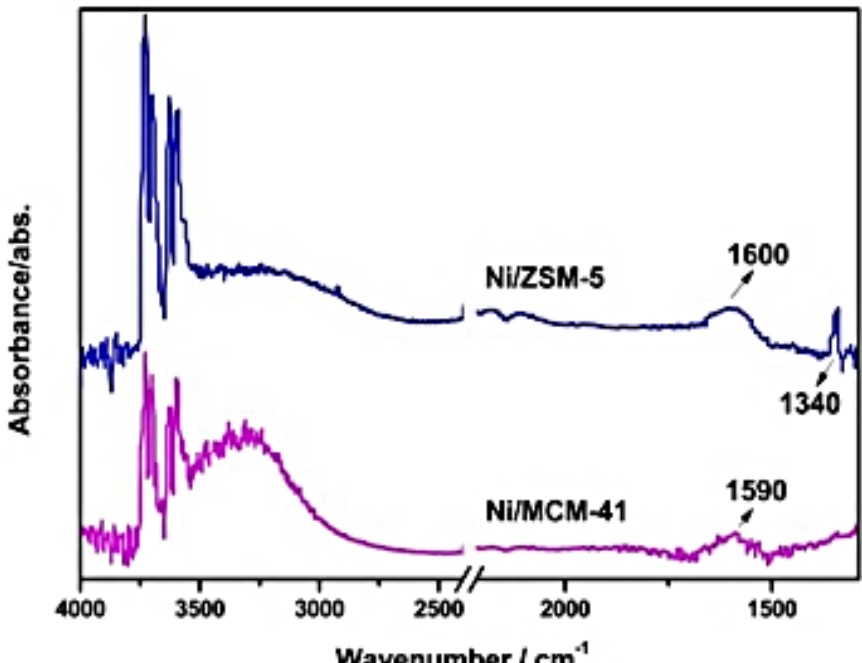

**Figure 20.** FT–IR spectra of adsorbed species on Ni/ZSM−5 and Ni/MCM−41 catalysts at 300 °C [61].

Diffuse reflectance infrared Fourier transform spectroscopy has been used to study the mechanisms of methanation reactions over Ni/ZSM−5 and Ni/MCM−41 catalysts (Figure 21). It was observed that N/S-24-Hy exhibited more catalytic activity in CO$_2$ hydrogenation than did N/S-Im catalysts in identical reaction conditions. The reaction pathway for hydrothermal treatment catalysts (N/S-24-Hy) exhibited a peak of methylene groups at low temperatures (250−500 °C), while the C-H peak of impregnation treatment (N/S−Im) catalysts appeared at above 300 °C. A high percentage of adsorbed CO$_2$ transformed into carbonate (1324 and 1432 cm$^{-1}$) species, as shown in Figure 21. In the case of the N/S−24Hy catalyst, the formation of formate species was observed due to a CO$_2$ reaction with hydrogen radicals, which is a critical intermediate for methane production [47].

Another reported study proved that the Ni−SiO$_2$ interface is responsible for CO$_2$ adsorption and activation. The availability of hydrogen coverage on the metal determines the direction of the intermediates (carbonate and formate) conversion toward CH$_4$ formation. Their improved combustion technique achieved a remarkable reactivity at low temperatures due to its high Ni dispersion and the interface between the metal and the support, as shown in Figure 22 [55].

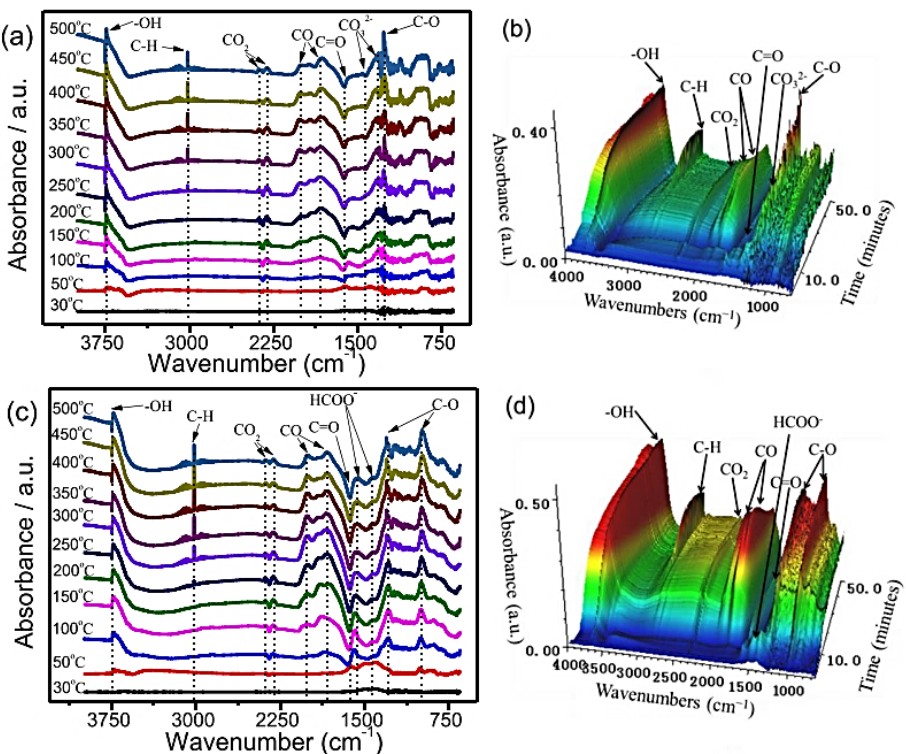

**Figure 21.** DRIFTS analysis of $CO_2$ methanation over (**a**,**b**) N/S−Im and (**c**,**d**) N/S−24-Hy catalysts [47].

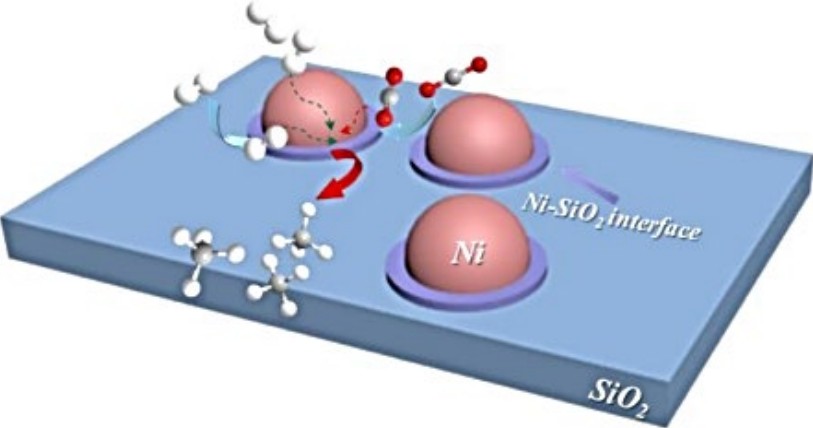

**Figure 22.** Activation of $CO_2$ and methanation reaction process over Ni-SiO$_2$ interface [55].

It has been observed that small metal particles lead to an increase in activity while big Ni particles produce CO as a secondary product. The generated defect site, which is associated with a lower Ni particle size, enhances the adsorption and activation of $CO_2$ and the overall process of $CH_4$ production. Additionally, small Ni particles prevent CO formation as secondary product. Pathway 1, as shown in Figure 23, involves the formation of monodentate carbonate and is considered the most effective route for $CO_2$ methanation [20]. However, when promoters such as $CeO_2$ surround Ni nanoparticles (Figure 24c), they cause excellent Ni dispersion by limiting agglomeration (Figure 24a) and increasing the amount of hydrogen activation sites on the Ni metal (Figure 24b). This phenomenon enhances catalytic methanation activity; however, increasing the $CeO_2$ concentration to 30 wt.% reduces the number of active sites and catalyst's performance. The effects of these metals, supports, and additives result in excellent metal dispersion and $CO_2$ adsorption. It has been reported that 20 wt.% of $CeO_2$ is the optimum loading for high methanation activity [70].

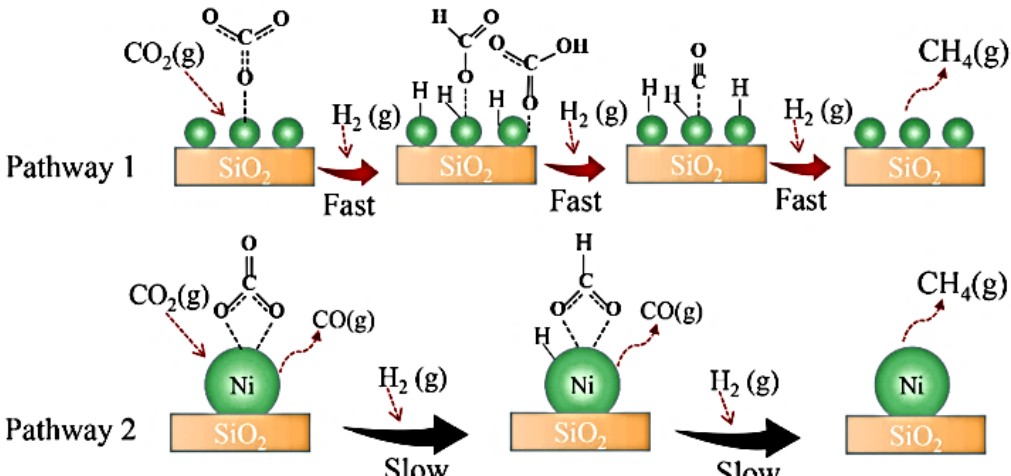

**Figure 23.** Suggested reaction mechanism for $CO_2$ methanation on catalysts with different Ni metal particle sizes [20].

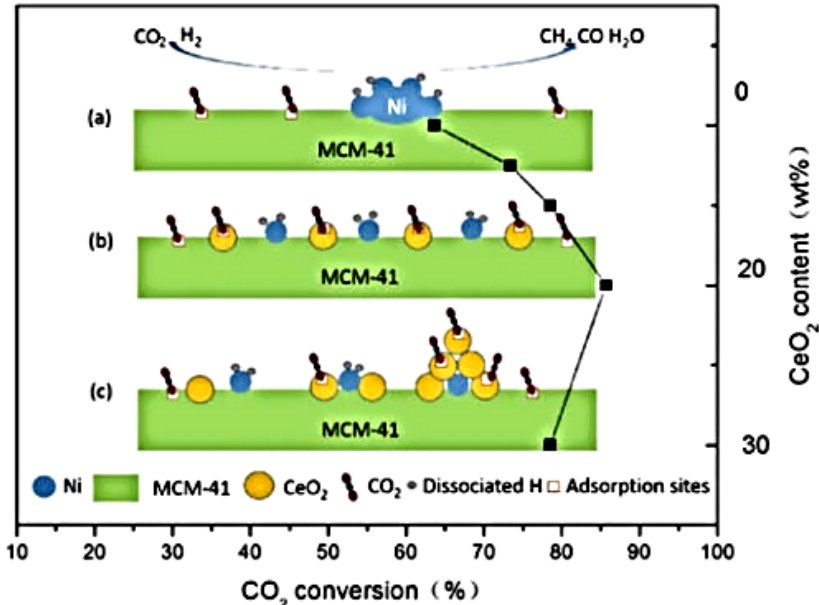

**Figure 24.** Mechanism of $CeO_2-Ni/MCM-41$ catalysts [70].

In another report, researchers proposed two possible paths for the $CO_2$ methanation process over nickel phyllosilicate (NiPs-x) catalysts (Figure 25). The $CO_2$ molecule interacts with the OH group and Lewis acid site on the NiPS surface to create intermediate carbonates (polydentate, monodentate, and bicarbonate). Then, the carbonates convert into formate species by reacting with the hydrogen that was spread on the Ni metal particles. Finally, the CO formed due to formate decomposition converts into methane. In the second path, the direct decomposition of $CO_2$ on the Ni metal is transformed into $CH_4$ by the hydrogen [38]. As shown in Figure 26, the Ni sites on the surface of an MSN support help decompose $H_2$ molecules into hydrogen atoms and assist in formation of oxygen vacancies (Figure 25 (A)). The formed oxygen vacancies activate $CO_2$ to form carbon species, which further react with hydrogen atoms to produce methane [71]. From these studies, it can be concluded that Ni is vital for the formation of the hydrogen radicals that are essential in the conversion of $CO_2$ into $CH_4$. Therefore, designers of catalysts should aim to create small particles of Ni to enhance reaction reactivity. This can be achieved using additives or reduction during pretreatment.

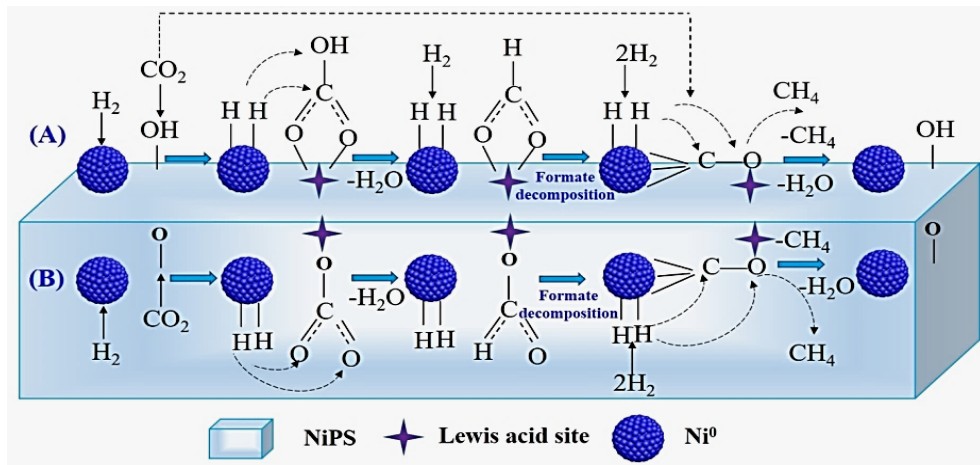

**Figure 25.** (A,B) Proposed processes of $CO_2$ methanation reaction over the NiPS catalysts [38].

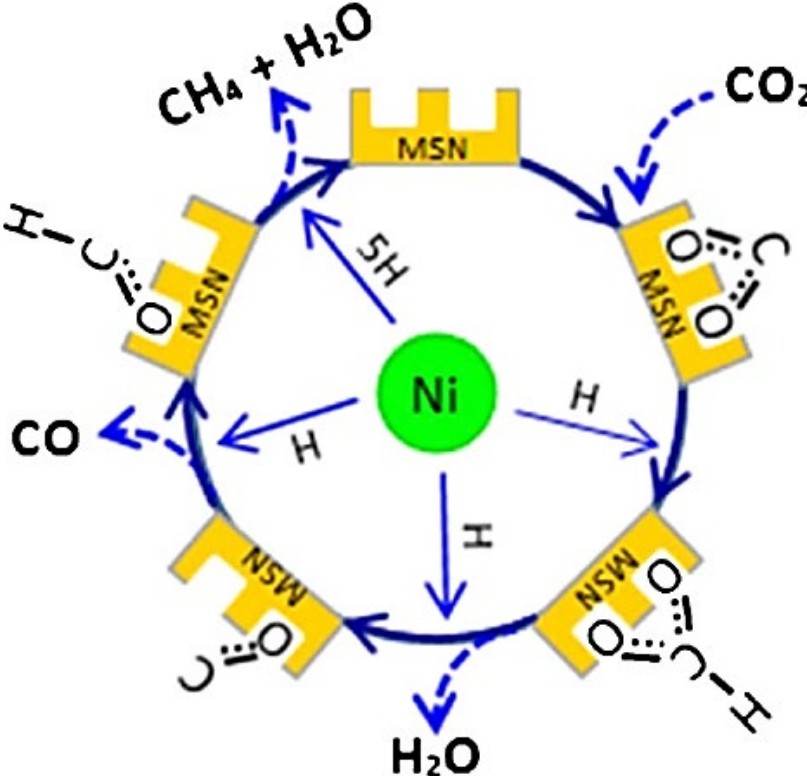

**Figure 26.** A possible mechanism of $CO_2$ conversion into $CH_4$ over Ni/MSN catalysts [71].

## 14. Effect of the Dopant

The development of Mn-promoted Ni catalysts has contributed to reaching elevated levels of metal dispersion and $CO_2$ adsorption and achieving superior performance in CO and $CO_2$ methanation [11]. Paviotti et al. used rice husks as a source for silica to form Ni-Ru supported by $SiO_2$ catalysts via the wet impregnation method for a $CO_2$ methanation reaction at temperatures ranging between 250 and 400 °C. The ratio of Ru/Ni was kept constant at 0.1 during the alteration of the metal loading. The optimum catalyst for the reaction of $CO_2$ conversion to methane, with improved stability, selectivity, and activity, was found to be a rice husk-derived silica with a ratio of 5 wt.% of Ni and 0.8 wt.% of Ru. The addition of a low amount of Ru resulted in a lower particle size and high ability to adsorb $H_2$ [72].

In another study, LaNiCo-based catalysts were tested to study the effects of bimetallic additives on catalytic activity at low temperatures and on stability at high temperatures. As shown in Figure 27, the $LaNi_{1-x}Co_xO_3$ catalyst exhibited a more remarkable reactivity at lower temperatures than did the monometallic catalyst. The optimum percentage ratio of the $Co/(Co + Ni)$ molar to increase activity was found to be (5%). The bimetallic active center led to a decrease in the activation energy of the $H_2$ and $CO_2$. Out of all the prepared catalysts, $LaNi_{0.95}Co_{0.05}O_3/MCF$ showed significant activity as a result of its nanosized particles of Ni and Co and the combined characteristics of both metals. It proved to have stable abilities for 100 h with no noticeable sintering [73]. Riani et al. estimated the effects of introducing lanthanum and silica to Ni-based catalysts supported by alumina for $CO_2$ methanation. The introduction of the silica prevents the formation of a perovskite structure, protects the support form, and enables the elevation loading of the $La_2O_3$, leading to the monitoring of acid-base characteristics. However, introducing Si to the support material $Al_2O_3$ lowers the Ni catalytic activity in $CO_2$ methanation due to the lower $CO_2$ adsorption. The catalysts were synthesized with a constant amount of 13.5 wt.% of Ni and the $La_2O_3$ loadings varied, which improved the stability and achievement of a distinctive $CO_2$ adsorption. The optimum amount of lanthanum on the $Ni/SiO_2$-$\gamma$-$Al_2O_3$ catalyst was equal to 37 wt.%, which increased the methane production to 83% and the selectivity to around 100% at low temperatures of about 300 °C [74].

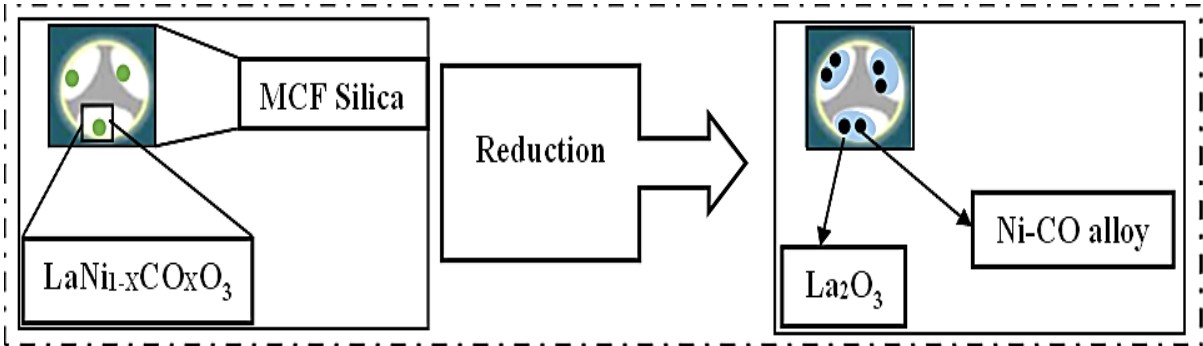

**Figure 27.** Fabrication of the $LaNi_{1-x}Co_xO_3/MCF$ catalyst [73]. (Copyright. Elsevier).

Taherian et al. investigated the addition of yttria to nickel-based catalysts supported on modified magnesia MCM-41 by preparing samples comprising differing yttria content in $xY_2O_3-Ni/MgO-MCM-41$ catalysts via the direct fabrication strategy for a Sabatier reaction. The authors concluded that the samples employing yttria gained significant activity compared to the unemployed samples. The greatest conversion of 65.55% and selectivity of 84.44% at 673 K (398 °C) were from a sample containing 2 wt.% of yttria due to the small size of the particles and the well dispersed Ni. Additionally, these rates lasted for the reaction duration of 30 h [75]. Guo and Lu reported the promotional impact of different alkaline-earth metal oxides on consistent $Ni/MO/SiO_2$ catalysts, where M refers to (MgO, CaO, SrO, and BaO) synthesized via the sequential impregnation route for $CO_2$ hydrogenation. Based on their analyses, the introduction of SrO to the catalyst increased its activity and stability. In terms of BaO promotion, while the activity was improved, the $Ni/BaO/SiO_2$ only lasted for 50 h due to destruction caused by Ni deactivation. Introducing CaO to the catalyst slightly influenced the efficiency of the $Ni/CaO/SiO_2$, and the addition of MgO remarkably obstructed its methanation productivity due to the diminished reducibility of Ni [76].

Vrijburg et al. studied the effects of a Mn promoter on $Ni/SiO_2$-$Al_2O_3$ catalysts in the Sabatier reaction, where the atomic ratios of Mn/Ni varied from 0 to 0.25. According to the reported results, the activity of the reaction can be enhanced by increasing the Mn/Ni ratio. At low temperatures, the catalysts show high stability, Ni dispersion, and methane selectivity. Based on IR spectroscopy results, the addition of Mn caused the adsorption and

activation of carbon dioxide at low temperatures [77]. In another study, to investigate the effects of the $CeO_2$ promoter, a group of $Ni_{1-x}CeO_2$/MCM-41 catalysts were prepared with 20 wt.% of nickel using the deposition precipitation method for a Sabatier reaction. The catalysts promoted with $CeO_2$ exhibited an increased reaction reactivity in comparison to the unpromoted Ni/MCM-41 catalyst. The 20 wt.% $CeO_2$ catalyst achieved a superior catalytic activity, with 85.6% $CO_2$ conversion and 99.8% methane selectivity at 380 °C. The addition of $CeO_2$ led to enhancements in the dispersed Ni, $CO_2$ adsorption, and overall efficiency of the reaction, which is the result of the combined characteristics of the metal, support, and promoter. Additionally, this catalyst remained stable for 30 h [70]. The catalytic activity of Ru nanoparticles on a nickel-based silica catalyst has been investigated during methanation. This preparation approach formed unalloyed metals on a layer of oxide passivation. This catalyst proved its validity by gaining 100% conversion at the low temperature of 200 °C with a 940 $h^{-1}$ TOF. Treatment with hydrogen can activate a catalyst again after coke formation during the reaction. By testing different nanoparticles such as Re, Rh, Ir, or Pd/Ni, the authors showed that the Re/Ni catalyst offers superior activity with high methane production at 460 °C and a TOF of 13,855 $h^{-1}$ [24].

The effects of adding rare earth metals (which have potential to replace noble metals) to nickel silica catalysts for a Sabatier reaction have been studied. Bimetallic catalysts were synthesized via the impregnation route using silica formed from electrospinning to enlarge the surface area. Among the additives of bimetallic lanthanide (La, Ce, Pr, Sm, Dy, and Yb), the bimetallic oxides praseodymium and cerium exhibited the most remarkable catalytic activity, increasing the activity of their catalysts to four times that of the $NiO/CeO_2$ catalyst and ten times that of the 5 wt.% $Rh/Al_2O_3$ catalyst. This activity is due to the synergic effect of the nickel and the 4f block of the rare earth elements that affects the base, reduces the size of the particles, and enhances the reaction's lifetime [78]. A core of self-assembled nickel nanoparticles with ceria nanowires (Ni-$CeO_2$ NWs) in shelled, microporous $SiO_2$ was fabricated using the one-pot method. This designed Ni-$CeO_2$@$SiO_2$ catalyst with a diameter of 2.9 nm exhibited excellent performance and stability in the conversion of $CO_2$ to $CH_4$. This is due to several factors, such as its distinctive texture, its intensive interaction, and the combined influence of Ni nanoparticles and $CeO_2$ nanowires in the presence of oxygen vacancies [79]. Li et al. prepared Ni-MgO nanoparticles supported by a core-shell silica for a Sabatier reaction at low temperatures with differing ratios of Ni/Mg. The optimum performance, with 87% activity, 99% selectivity and 100 h of stability at 300 °C, was from the Ni/Mg ratio equal to 4/1. This is due to the separation of dispersed Ni nanoparticles preserved by the silica shell [80].

Additionally, Guo and Lu investigated the influence of varying ratios of Co/Ni, where a ratio of 0.4–1 exhibited an enhanced performance of the bimetallic catalysts in $CO_2$ methanation at temperatures between 250 and 350 °C [27]. In another study, an alloy Ni-Pd catalyst on SBA-15 as its support performed differently according to varying ratios of Ni-Pd. Bimetallic catalysts with a ratio of 3:1 gained the highest methane production, with 0.93 mol/$CO_2$ mole-formed methane at 430 °C [25]. The loading of boron on a fibrous-silica-nickel catalyst has been shown to increase catalyst efficiency in $CO_2$ methanation. The superior production rate was 84.3% in the reaction condition of 10,500 GHSV, 6 gas ratio $H_2/CO_2$, and 500 °C [81]. All the discussed scientific studies' perspectives regarding a catalyst's stability can be concluded in Figure 28.

Comparison of different Ni/pure-$SiO_2$ and Ni/modified-$SiO_2$ catalysts in $CO_2$ methanation are presented in Tables 1 and 2 respectively. The observations from the data indicate that Ni/modified $SiO_2$ catalysts show better performance compared to the Ni/pure $SiO_2$ catalysts due to the promoting effect of different promoters used to modify the support. Furthermore, the method of the preparation, morphology of $SiO_2$ and textural properties of the catalysts also shows significant impact on $CO_2$ conversion and selectivity to $CH_4$ production.

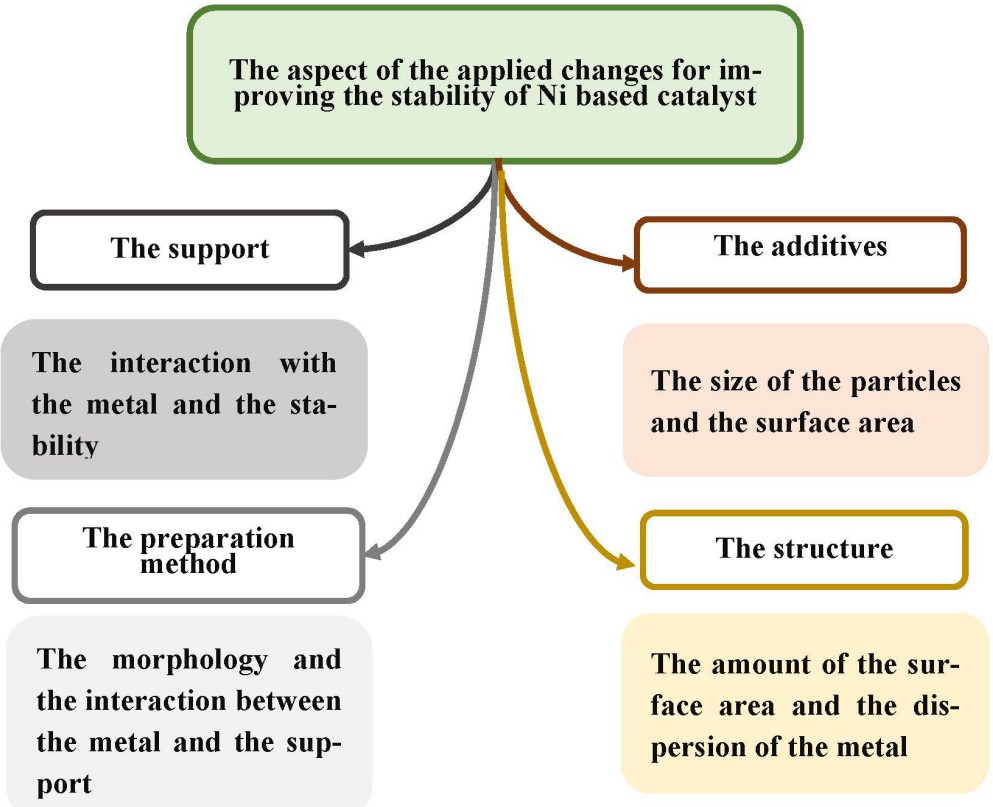

**Figure 28.** Summary of the different modifications to improve the stability of Ni-SiO$_2$-based catalysts.

**Table 1.** Comparison of different Ni/pure-SiO$_2$ catalysts in CO$_2$ methanation.

| Catalyst | Preparation Method | Reaction Conditions (GHSV (mL·g$^{-1}$·h$^{-1}$)) P (MPa) | Conv. CO$_2$ (%) | Select. CH$_4$ (%) | Stability (h) | Temp. (°C) | Rate (mol CH$_4$·g Ni$^{-1}$·h$^{-1}$) | Ref. |
|---|---|---|---|---|---|---|---|---|
| 10.4%Ni/SiO$_2$-C | One-pot sol–gel | 20,000, P 2.0 | 77.2 | 99.8 | 52 | 310 | 1.32 | [52] |
| Ni-I/m-SiO$_2$ | Wet impregnation | 23,100, P 0.19 | ~80 | ~100 | - | 400 | 1.65 | [59] |
| 40%Ni/SiO$_2$-IM | Impregnation | 30,000, P 0.1 | ~70 | ~95 | 60 | 370 | 0.15 | [53] |
| 40%Ni/SiO$_2$-AEM | Facile ammonia-evaporation | 30,000, P 0.1 | 82.4 | 95.5 | 60 | 370 | 0.27 | [53] |
| Ni/SiO$_2$ | Impregnation | 10,000, P 0.1 | ~55 | ~90 | 30 | 400 | 0.21 | [34] |
| 10 wt.%Ni-1 wt.%MgO/SiO$_2$ | Co-impregnation | 15,000, P 0.1 | ~67.0 | ~98.0 | 50 | 350 | 0.88 | [58] |
| 10%Ni/SiO$_2$ | Impregnation | 60,000 | 53.5 | 96 | 5 | 350 | - | [69] |
| 10%Ni/SiO$_2$-Gly | Combustion–impregnation | 30,000 | 66.9 | 94.1 | 50 | 350 | - | [55] |
| 10%Ni/SiO$_2$-Cit | Combustion–impregnation | 30,000 | 39.1 | 84.9 | 50 | 300 | - | [55] |
| N/S-24-Hy | Incipient wetness impregnation | 60,000, P 0.1 | 80.4 | 99.8 | 100 | 400 | - | [47] |
| 2 wt.% Ni/SiO$_2$-ED | Impregnation with [Ni(EDTA)]$^{2-}$ | 12,000 | 80 | 90 | - | 450 | - | [20] |
| Ni5Ru/SiO$_2$ | Wet impregnation | 6000 | 71.0 | 92 | 24 | 400 | - | [72] |
| 10Ni/SiO$_2$ | Sequential impregnation | 15,000 | 73.2 | 98.7 | 50 | 400 | - | [76] |
| Ni/Mg/Si | Sequential impregnation | 15,000 | 61.9 | 92.1 | 50 | 400 | - | [76] |
| Ni/Ca/Si | Sequential impregnation | 15,000 | 73.3 | 98.9 | 50 | 400 | - | [76] |
| Ni/Sr/Si | Sequential impregnation | 15,000 | 76.3 | 99.0 | 50 | 400 | - | [76] |

**Table 1.** *Cont.*

| Catalyst | Preparation Method | Reaction Conditions (GHSV (mL·g$^{-1}$·h$^{-1}$)) P (MPa) | Conv. CO$_2$ (%) | Select. CH$_4$ (%) | Stability (h) | Temp. (°C) | Rate (mol CH$_4$·g Ni$^{-1}$·h$^{-1}$) | Ref. |
|---|---|---|---|---|---|---|---|---|
| Ni/Ba/Si | Sequential impregnation | 15,000 | 74.9 | 98.9 | 50 | 400 | - | [76] |
| 5%Ni/SiO$_2$ | Wet impregnation | 60,000 | 52.7 | 92.8 | 5 | 400 | - | [69] |
| 10%Ni/SiO$_2$ | Wet impregnation | 60,000 | 53.5 | 96 | 5 | 350 | - | [69] |
| 15%Ni/SiO$_2$ | Wet impregnation | 60,000 | 55 | 96.1 | 5 | 350 | - | [69] |
| 10%Ni-1%Cu/SiO$_2$ | Wet impregnation | 60,000 | 39.5 | 44.4 | 5 | 400 | - | [69] |
| 10%Ni/SiO$_2$ | Impregnation | 2400 | 68 | 66 | 100 | 400 | - | [61] |
| 34.3 wt.% NiPS-1.6 | Hydrothermal | 40,000 | >80 | ~100 | 48 | 330 | - | [38] |
| 22.6%N180/SR-U-24 | Hydrothermal reaction | 60,000 | 67.5 | - | 100 | 450 | - | [50] |
| Ni$_{0.8}$Mg$_{0.2}$O@SiO$_2$ | Chemical co-precipitation and modified Stöber synthesis | 60,000 | 87.0 | 99.0 | 100 | 300 | 86.1 | [80] |
| Co$_{0.4}$Ni/SiO$_2$ | Wet co-impregnation | 13,200 | 83.2 | 82.4 | - | 350 | - | [27] |

**Table 2.** Comparison of different Ni/modified-SiO$_2$ catalysts in CO$_2$ methanation.

| Catalyst | Preparation Method | Reaction Conditions (GHSV (mL·g$^{-1}$·h$^{-1}$) P (MPa)) | Conv. of CO$_2$ (%) | Select. to CH$_4$ (%) | Stability (h) | Temp. (°C) | Rate (mol CH$_4$·g Ni$^{-1}$·h$^{-1}$) | Ref. |
|---|---|---|---|---|---|---|---|---|
| Ni/SiO$_2$-ZrO$_2$ | Impregnation | 10,000, P 0.1 | ~75 | 100 | 30 | 400 | 0.32 | [34] |
| Ni/SiO$_2$-Al$_2$O$_3$ | Impregnation | 10,000, P 0.1 | ~70 | 100 | 30 | 400 | 0.30 | [34] |
| Ni-SiO$_2$/GO-Ni-foam | Hydrothermal | 500, P 0.1 | ~80 | ~91 | 72 | 470 | - | [63] |
| 10%Ni-La$_2$O$_3$/SBA-15 | Impregnation | 6000 | 54 | 99 | 160 | 320 | - | [48] |
| 10%Ni-La$_2$O$_3$/SBA-15(C) | Citrate complexation | 6000 | 90.7 | 99.5 | 160 | 320 | - | [48] |
| Ni/SBA-15-Op | Facile one-pot hydrothermal | 10,000, P 0.1 | ~76 | ~97 | 50 | 420 | - | [45] |
| Ni/LaSi | Successive impregnation | 4650 | 83 | 98 | 6 | 250 | - | [8] |
| 5 wt % Ni/MSN | Wet impregnation | 50,000 | 64.1 | 99.9 | 200 | 300 | - | [51] |
| LaNi$_{0.95}$Co$_{0.05}$O$_3$/MCF | Citric acid-assisted impregnation | 60,000, P 0.1 | 75.4 | ~95 | 100 | 450 | - | [73] |
| Ni@HZSM-5 | Hydrothermal synthesis | | 66.2 | 99.8 | 40 | 400 | - | [56] |
| Ni/ZSM-5 | Impregnation | 2400 | 76 | 99 | 100 | 400 | - | [61] |
| Ni/MCM-41 | Incipient wetness impregnation | 48 | 60 | 90 | | 360 | - | [57] |
| Ni-MCM-41 | One-pot synthesis | 19,700 | 20 | 80 | | 400 | - | [43] |
| 5%Ni/MCF_ch_iwi | Incipient wetness impregnation | 8600 | 62 | 97 | 20 | 350 | - | [60] |
| 5%Ni/MCF_ch_iwi | Incipient wetness impregnation | 8600 | 77 | 94 | 20 | 400 | - | [60] |
| 5% Ni/MCM_iwi | Incipient wetness impregnation | 8600 | 44 | 78 | 20 | 350 | - | [60] |
| 5% Ni/MCM_iwi | Incipient wetness impregnation | 8600 | 70 | 95 | 20 | 400 | - | [60] |
| Ni-MCF | One-pot synthesis | 8600 | 39 | 58 | 20 | 400 | - | [60] |
| Ni-MCM | One-pot synthesis | 8600 | 51 | 79 | 20 | 400 | - | [60] |
| 30Ni/Al$_2$O$_3$.0.5SiO$_2$ | Sol–gel synthesis | 12,000 | 82.4 | 98.2 | 30 | 350 | - | [66] |
| Ni/MSN | Impregnation | 50,000 | 85.4 | 99.9 | 0.8 | 350 | - | [46] |
| 5%Ni/fibrous SBA-15 | Incipient wetness impregnation | 24,900 | 98.9 | 99.6 | 120 | 400 | - | [36] |
| Ni$_{0.75}$Pd$_{0.25}$/SBA-15 | One-pot wet chemical and impregnation | 6000 | 96.1 | 93.7 | - | 430 | 0.93 | [25] |
| 0.5B-FSN | Wet impregnation | 13,500 | - | - | - | 525 | - | [81] |
| 5wt.% Ni/MCM-41 | Wet impregnation | 50,000 | 56.5 | 98.3 | 200 | 300 | - | [51] |
| 20 wt.%Ni-CeO$_2$/MCM-41 | Deposition–precipitation | 9000 | 85.6 | 99.8 | 30 | 380 | - | [70] |
| Ni/MCM-41 | Deposition–precipitation | 9000 | 64 | 96 | 30 | 380 | - | [70] |
| LaNiO$_3$/MCF(30LMON-C-650) | Citric acid-assisted impregnation | 60,000, P 0.1 | 76 | 97 | 100 | 450 | - | [64] |

## 15. Theoretical Studies

It is widely approved that the combination of experimental and theoretical studies helps researchers understand the different pathways of $CO_2$ activation and how C–C coupling proceeds on a Ni particle surface. Theoretical studies allow researchers to not only tune the activity and selectivity of the $CO_2$ methanation process but also gain fundamental knowledge about some basic principles of heterogeneous catalysis on different metals, including Ni-based catalysts. Vogt et al. performed a systematic study to fully catalog and identify the experimental and theoretical activity and selectivity descriptors for catalytic $CO_2$ methanation processes over Ni catalysts. These authors used computational catalysis (density functional theory, DFT) to understand the basic concepts of $CO_2$ methanation over Ni-supported catalysts. Through calculations, for example, the application of microkinetic modeling, a detailed study of surface reactions at the molecular level was used [82].

Density functional theory enables the exploration of electronic and geometric characteristics of heterogenous catalysts. To control the production selectivity of $CO_2$ conversion, the design and fabrication of Ni/silica-based catalysts via theoretical methods is an effective approach [82]. According to DFT calculations, the movement of an electron between energy levels of the orbitals causes $CO_2$ activation along with other factors, such as oxygen vacancies and Lewis acid sites on the surfaces of catalysts [3]. It has been found that $CO_2$ hydrogenation over Ni catalysts can and does produce $C_3H_8$, and that activity and selectivity can be tuned by supporting different Ni particle sizes on various metal oxides, such as $CeO_2$ and $TiO_2$. Therefore, theoretical studies are not only useful for the highly selective production of $CH_4$, but they can also provide new insights for $CO_2$ activation and subsequent C–C coupling towards value-added products.

## 16. Conclusions

Producing energy from fossil fuel resources results in the earth's atmosphere being filled with massive greenhouse gas emissions and air pollution. Converting $CO_2$ into value-added hydrocarbons via environmentally friendly processes is one the best alternatives to counter greenhouse gas emissions. Ni and $SiO_2$-based catalysts are commonly used materials due to their feasibility and functionality for the conversion of $CO_2$ to hydrocarbons. Catalyst modifications can be performed on different aspects of catalysts to overcome the thermal limitations of $CO_2$ methanation. To ensure nickel-based catalysts are sinter resistant and active at low temperatures, their structure, Ni metal particle size, dispersion, and interaction between the Ni and $SiO_2$ support should be taken into consideration during the design and fabrication of Ni-$SiO_2$-based methanation catalysts. In this review, we attempted to summarize recent developments in the preparation technology, pretreatment, reaction conditions, loading amount, and support nature of Ni-based catalysts, and the dopants used to increase the catalytic activity and stability of the methanation reaction were discussed. Novel preparation methods provide a higher performance compared to the traditional ones. Doping with rare earth metals and transition metals enhances Ni metal dispersion and the number of active centers via a catalyst's basicity, resulting in better activity and stability. Modified $SiO_2$ supports improve a catalysts' structure and its catalytic activity in $CO_2$ methanation reactions. The effects of experimental conditions and the mechanisms of Ni-based catalysts have been discovered for $CO_2$ methanation reactions. In addition, theoretical studies allow researchers to not only tune the activity and selectivity of the $CO_2$ methanation process but also gain fundamental knowledge about some basic principles of heterogeneous catalysis on different metals, including Ni-based catalysts.

## 17. Future Perspective

The design and fabrication of highly stable and low temperature-active Ni-$SiO_2$-based methanation catalysts could be a major potential improvement in the future. Additionally, the nature of a support material can be adjusted to optimize its interaction with active Ni metal nanoparticles and prevent metal sintering. Studying the reaction mechanisms of $CO_2$ conversion to methane or higher hydrocarbons in detail will enable advanced development.

Finding a suitable, reproducible preparation method for the synthesis of Ni-SiO$_2$-based catalysts could be cost effective and a critical criterion for industry applications.

**Author Contributions:** N.A. carried out all the literature reports, analyzed the data and the mainly responsible for writing-original draft of the paper; K.N. and Q.A.A. supervised review writing and editing; K.N. designed the review; K.N. and Q.A.A. corrected the review. All authors have read and agreed to the published version of the manuscript.

**Funding:** This research received no external funding.

**Data Availability Statement:** Data analyzed in this review were a re-analysis of existing literature, which are openly available at journals websites cited in the reference section.

**Conflicts of Interest:** The authors declare no conflict of interest.

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
