# Peer review of "Recent Progress in Nickel and Silica Containing Catalysts for CO2 Hydrogenation to CH4"

_catalysts, doi:10.3390/catal13071104_

Round 1

Reviewer 1 Report

The review article reviews the literature on the CO2 methanation process catalyzed by Ni and SiO2 based catalysts, in particular the silica supported Ni metal, and points to the future research directions.

The topic of CCU is of significance and the scope is adequate.

First and foremost, the heading numbering system is problematic: 1. Introduction, but Conclusions and Future perspective are not numbered. 15 headings are a lot and these are not well organized. 11 and 12 can be sub-headings of 10.

More editing is needed. Some figures are overly large and outside the page. Fig, 6, 7 are missing. Some text can not be seen in the text boxes in many figures.

English usage is poor which compromises the flow of the review. Thus a thorough check is necessary by a native English speaker.  

In conclusion, a major revision is necessary.

Some more for consideration:

1. CCUS rather than CCS should be introduced. CCUS is carbon capture, utilization and storage. Also because the review content belongs to utilization.

2. Are figures all remade from the references? Any figure directly taken from the literature needs permission. And the ones without references are original?

3. In the introduction, the early theoretical work explaining CO2 hydrogenation mechanism should be cited. doi.org/10.1021/jp8048586

How about other methods besides CO2 hydrogenation? To make the Introduction comprehensive, consider including converting CO2 to oxazolidinones at room temperature doi.org/10.1021/jp801348w

Extensive revision

Author Response

Responses to reviewer comments file is attached

Reviewer 2 Report

The authors discussed the recent application of Ni and SiO2-based catalysts for CO2 methanation, especially the previously reported research on silica supported Ni metal. It provides guidance for the design of CO2 hydrogenation catalysts. However, a review should summarize the critical assessment and opinions of the relevant literature rather than simply describe the experimental results of the article. In this review, the authors summarize the literature while also need to incorporating their perspective and analysis. Therefore, the manuscript still has plenty of room for improvement, the following are some issues that should be addressed by the authors before publication:

1. All the figures in the review need to be adjusted, and the contents of many figures are not fully displayed. In addition, figures 6 and 7 are missing.

2. There are some published reviews of Ni-based catalysts for CO2 hydrogenation. What is the difference between this article and other existing reviews in the field? It could be illustrated at suitable positions in the manuscript.

3. In addition to the influence of different Ni-based catalysts on the activity and stability of CO2 hydrogenation reaction, the authors also should discuss in detail the mechanistic pathway and reaction thermodynamics of CO2 hydrogenation. Understanding the catalytic reaction mechanism is an important basis for the design and preparation of efficient catalysts.

4. There is also a large amount of work related to theoretical calculations in the Ni-based catalysts for CO2 hydrogenation to CH4, which is not discussed in this review.

Author Response

(The authors gave the same response as above.)

Round 2

Reviewer 1 Report

The authors have addressed the issues rasied. It is recommended for publication.

Reviewer 2 Report

The authors have revised the manuscript according to the comments and suggestions from Reviewers. It could be accepted for publication.